# Advanced monitoring and numerical modelling of the stability, safety and reliability indicators of the earthen dam of Songloulou (Cameroon)

Zoa Ambassa[1,2]*, Jean Chills Amba[1,2], Merlin Bodol Momha[1], Landry Djopkop Kouanang[1,2], Robert Nzengwa[1,2], Pascal Adrien Mbongo[3]

1 Laboratory of Energy Modelling Materials and Methods (E3M), National Higher Polytechnic School of Douala, University of Douala, Douala, Cameroon, 2 Department of Civil Engineering, National Higher Polytechnic School of Douala, University of Douala, Douala, Cameroon, 3 The Energy of Cameroon (ENEO), Douala, Cameroon

* daniel.zoa77@yahoo.fr

**Data Availability Statement:** All data files are available from the figshare at 10.6084/m9.figshare.23936247

## Abstract

For the determination of global stability after long term advanced monitoring, artificial intelligence have been used for the data analysis of water level and displacements of Songloulou earth dam at Cameroon. Measurements of safety and reliability indicators follow changes set by piezometric and pendulums measurements. The results obtained from the artificial intelligence on the base of many years recording data have confirmed the relevance and robustness of this model. The ANFIS model combining the concept of neural network and fuzzy logic was used to simulate the behaviour of piezometers and pendulums in the dam. This model has provided satisfactory results, given in the large amount of data to be processed. The water level evolution is modelled using the ANFIS function integrated in the MATLAB software and the result is compared to that obtained by the HST method. Afterwards, the state of stress on the structure and stability of the slope at shear have been assessed based on the hydro mechanical behaviour using the GEOSTUDIO Finite Element computation software. The input parameters are: the head of water recorded in the piezometers and geotechnical parameters of the dam. The modelling results in terms of displacement are accurately consistent with the displacement measurements. The horizontal displacement of pendulums obtained by GEOSTUDIO is 80 mm and those measured directly of the pendulums have 70 mm of average value. The safety factor for slope stability according to 530 m water level is 1.5.

## 1. Introduction

The evaluation and analysis of the monitoring data of a dam during its lifetime make it possible to process the influence of external parameters on these measurements [1–4]. The dam's recorded measures variation results of a combination of correlated factors. Three factors are

**Funding:** The authors received no specific funding for this work.

**Competing interests:** The authors have declared that no competing interests exist.

preponderant and two of them, the hydrostatic conditions i.e. the water level in the reservoir and the climatic conditions (temperature, rain, and slope sheet), are reversible if the stability limits or resistance are not reached [5–9]. The third factor is irreversible and related to the age of the dam [5–9]. An in-depth analysis of the structure assumes that we can follow the temporal evolution of its behaviour [1–9]. The first stage of the analysis consists in displaying the measurements of each instrument on a graph adapted to the phenomenon to be analyzed. In general, the graph is multiannual for mechanical quantities, depending on the dimension of the retention for hydraulic quantities [5–14]. This first level of analysis most often makes it possible to detect, at least qualitatively, sudden or slow irreversible variations, in particular when there are few explanatory factors (restraint at an almost constant level, measurements not subject to seasonal influences). However, this level of analysis is insufficient for large dams when numerical values are sought or when several explanatory factors coexist. In these cases, more sophisticated statistical models are used. For this category of structures, the importance of rapid processing of monitoring data considered essential for the analysis of overall or partial stability of the dam is obvious [15–17]. It is advisable to use deterministic models or even statistical models to improve the reliability of the measurements.

For deterministic models, finite element calculation models are commonly used [15]. Indeed, subject to a good knowledge of the parameters of material behaviour laws, these models make it possible to reliably represent strains and stresses in the embankment in different project situations. They also make it possible to assess the state of stress during the life of the structure, especially in the context of safety reassessment studies [18–25]. The limits of these approach lies in the fact that the calculations are quite laborious when one wishes to model the regular monitoring of the structure and, above all, they do not easily allow the consideration of the complexity of the factors which influence the nonlinear behaviour and the heterogeneities of soil, rock and concrete materials. However, the statistical analysis methods for monitoring data seems more advantageous because they allow to separate the respective influences of several explanatory factors introduced into the model. Among them, the 'HST' method which was originally developed for the pendulums of arch dams (Carrere) could be cited. From this method derives an approach such as the 'ACP' method which has the advantage of identifying the significant variables having a strong impact on the observed measurement. This method and its derivatives are used in several countries and their field of application has widened considerably.

On the other hand, for the particular case of hydraulic measurements (piezometry and flow rates), a hysteresis effect is commonly observed when representing the value of the pore pressure in the core or an earth dam foundation as a function of the water level in the reservoir. We can note that the path described when going up the body of water is not the same as the way down. Bonelli and collective [1, 6–8] explain this phenomenon by the fact that the soil capacity is never null, because the presence of dissolved or occluded air, even for compacted soils close to saturation. Thus, compared to the models described above, taking into account the delayed response seems more suitable for analysing data when there is a difference between the solicitation and the response measured by the monitoring instrument. Despite this consideration of the delayed response, it can be seen that the modelling of the evolution of the measured parameters does not match perfectly with the raw data. This delayed effect is more suitable for embankment dams but for concrete dams, the modelling seems more delicate given the number of parameters to be considered [6, 7].

In this paper, an analysis of data monitoring based on artificial neural networks is implemented. The method adapts well to the variations of parameters influencing the measurement of apparatus and does not require ongoing recalibration during the lifetime of the structure. Unlike the HST method based on data recorded during an analysis period, neural networks

use the entire data recorded during exploitation and improve their efficiency over time. Indeed, the neural network method allow to model nonlinearities based on the use of the properties of time series [26]. These nonlinearities are closely linked to the evolution of geotechnical parameters, environmental parameters, as well as structural parameters of the dam [27–29].

## 2. Presentation of structures and context of research

The environmental context and the characteristic of the structure are presented in this part. The same applies to the location of the monitoring devices of the dam.

### 2.1. Presentation of the Songloulou hydroelectric earthen dam

The Songloulou earthen dam is located in the Littoral region of Cameroon, towards Babimbi near Massok. It is a run and river dam implanted on the Sanaga River. With its vast watershed representing more than 20% of the total area of the country and its many rapids, the Sanaga is the largest river of Cameroon and constitutes a reservoir of hydroelectric energy of the first order. The main dam, levelled at height 530 m, and has a central core in compacted lateritic clay, separated from the upstream shell by a transition zone and separated to the downstream shell by a sand chimney filter. A riprap zone with quarry stones provides anti-swing protection on the upstream face and a 9.0 m wide crest. The spillway is calculated to evacuate an exceptional flood of 10,000 $m^3$/s below elevation 528.50 m with one blocked valve. It has seven passes closed by segment gates 14 m wide and 17 m high, controlled by winches and chains. The passes, whose sill is at elevation 511 m, are separated by pillars 4.5 m thick, which support the valves electromechanical equipment at elevation 530 m. The intake dam, with a maximum height of 35 m, is made up of eight valves 13.5 m wide separated by buttresses, a connecting gravity wall to the left bank and a connecting buttress structure at the spillway. The openings of the four intakes on the left bank are equipped with large screens, cleaned by a bar screen rolling on the crown of the structure at elevation 530.

### 2.2. Context and issue

Concrete swelling phenomena were identified six years after the commissioning of the first units (1981) by reducing the clearances between the fixed and moving parts of the turbines and alternators. These swellings had various consequences, making the operation and safety of the structures difficult. Studies have been undertaken since 1991, aimed at evaluating and improving the safety of the structures, as well as a series of interventions to counter the effects of swelling on the equipment with increased monitoring of the structures [30]. On the Songloulou's site, monitoring data is regularly collected but the site has no means of analysing this data automatically. The analysis of these monitoring data raises the questions about the effectiveness of the technique to be used given the advantages and limitations of monitoring techniques. The objective of this article is to analyse the monitoring data of the Songloulou hydroelectric dam using artificial intelligence and to propose a method of prediction and assistance in the analysis of these data for structure safe operation. The main advantages of the ANFIS model in practical engineering applications are the prediction the evolution of the critical phenomena as a function of time and environmental conditions and the failure mechanism of structures.

## 3. Materials and method of monitoring data analysis

The Songloulou hydroelectric dam has 12 pendulums to measure displacements, 40 piezometers for pressures and more than 200 other monitoring devices. In this paper we are interested

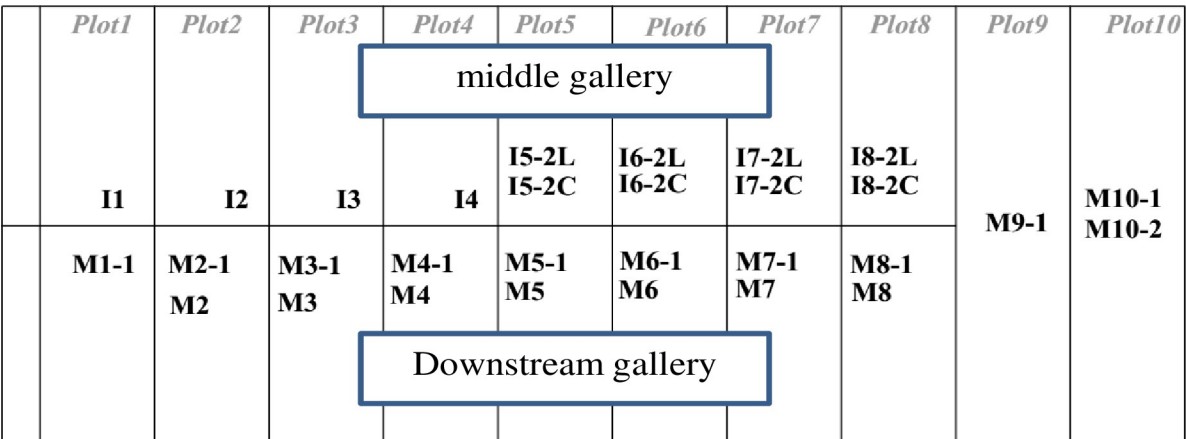

**Fig 1. Position of the piezometers at the Songloulou water intake dam.**

in the analysis of uplift pressures and horizontal displacements on the intake and the spillway dam. The layout of the piezometers in these works is illustrated in Figs 1–4.

The methodology consist to analyse this quantity of data collected on the site at regular times intervals for more than a decade using artificial intelligence model. Several research works have been carried out around the world using artificial intelligence to guarantee the reliability and safety of dams [26–37]. On the intake dam, 30 piezometers are usable out of the 31 installed. On the gallery spillway dam, 09 piezometers can be used. These two structures are constantly monitored. The range of data used for our study covers the period from 1997 to 2019 for the pendulums and from 2008 to 2019 for the piezometers.

## 3.1. Calibration and operation of the fuzzy neuron tool

For the computation of neural network, Matlab software as used [38]. Among the possibilities that this software offers in the "CONTROL SYSTEM DESIGN AND ANALYSIS" part, the "Neuro-fuzzy Designer" application, is the main tool of our programming. The model presented here use (Fig 5) three explanatory variables at the entry: the level of the reservoir (z), the season (s) and the time (t). The output value will represent the estimated behaviour of the piezometer.

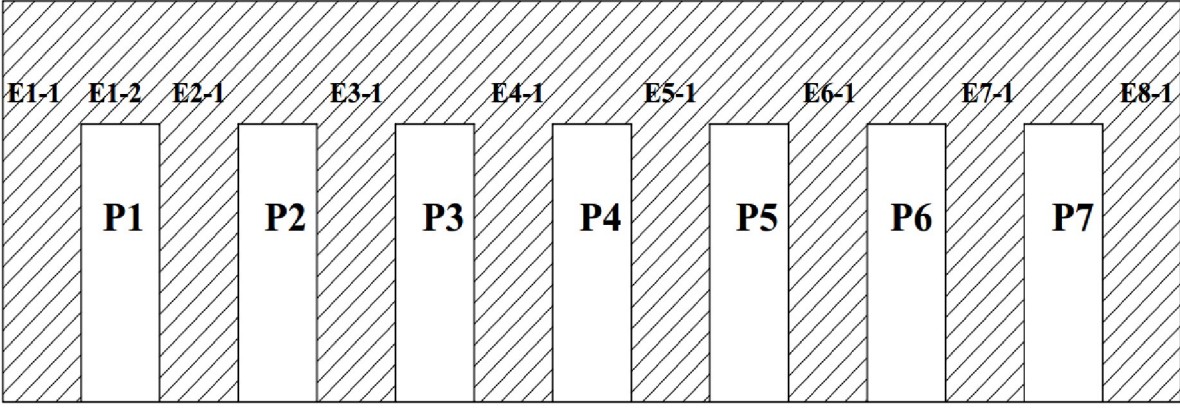

**Fig 2. Position of the piezometers at the Songloulou evacuator dam.**

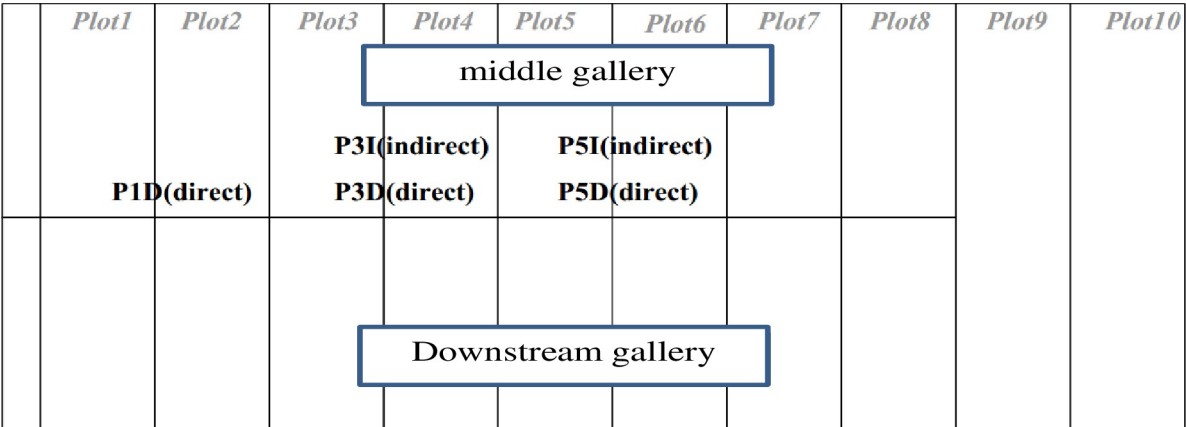

**Fig 3. Position of the pendulums at the intake dam of the Songloulou hydroelectric development.**

## 3.2. Network of the parameters of the fuzzy neuron method

All details used in this work are developed in the scientific and technical documents published by the aforementioned authors [9, 26, 28–37]. It should be recalled here that, the fuzzy neuron method is commonly used for the analysis of the behaviour of civil engineering and hydraulic structures. We refer to some recent works that have used this technique [38–45] with satisfactory results.

Table 1 describes the neural network developed. It can be interpreted as follows: at the entry we have explanatory variables which are here the level of the reservoir, the season and the time. The membership function is chosen to describe the phenomenon given that the curves are not affine and go jagged. We opted for the Bellshape function. The type of inference chosen in this study is Sugeno because the membership functions are precise values.

## 4. Analysis of the HTS model and that of the fuzzy neural network

### 4.1. Intake dam: Exploitation of piezometric data

**4.1.1. Use of piezometer I1.** Table 2 presents a summary of the new values of the membership function parameters of the piezometer I1 using Bellshape type membership functions. From the ANFIS model (Adaptive Neuro Fuzzy Inference System) the results of the piezometer I1 are given.

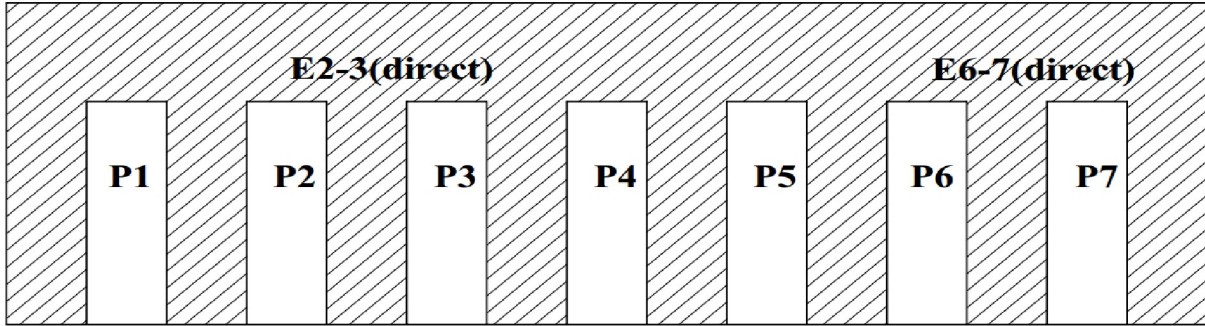

**Fig 4. Position of the pendulums at the spillway dam of the Songloulou hydroelectric facility.**

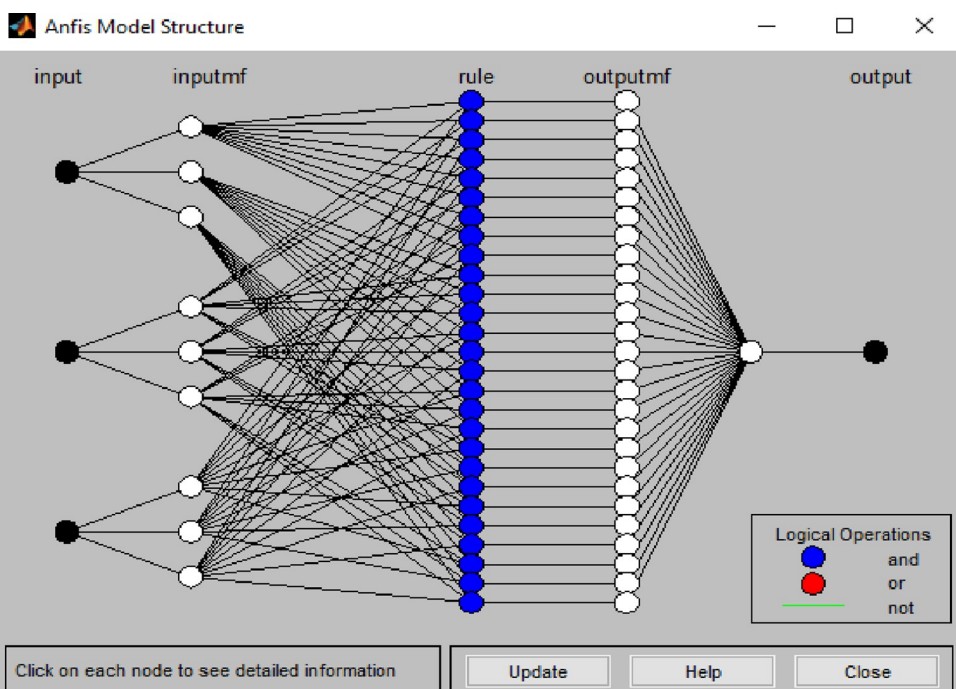

**Fig 5. Architecture of the ANFIS network in MATLAB software.**

After training and developing the new parameters, the final parameters $n_i$, $s_i$, $t_i$, and $b_i$ are determined according to the established rules (Table 3). Fig 6A shows that the season acts in sinusoidal way on the level of piezometer I1. This reflects periodic soil behaviour. The lower the water level in the dam, the lower the soil pressure. But a lag may exist between these variations due to the delay phenomenon. However at certain times, due to the variability of the seasons over time, there are slight differences at the level of the piezometer I1 measure. This effect proves that the soil reacts well to climatic hazards. The effects of water level and time on the level of this piezometer are highlighted in Fig 6B. This influence does not decrease over time, because the two preponderant variables are correlated. Over time, this should decrease due to the consolidation of the soil which should have led to the deletion of pore pressures.

Fig 7A–7C show that the ANFIS model is more compatible with the data observed on the I1 piezometer than the HST model which is unable to predict the peaks. It can be observed that

**Table 1. Presentation of the piezometer network.**

| | |
|---|---|
| **Number of entry** | **3** |
| Membership functions | Bellshape 3 for entrance |
| T-norm | Product |
| Type of inference | Sugeno |
| Number of neuron | 67 |
| Number of premise parameters | 27 = 3*3 |
| Number of consequent parameters | 108 = 27*4 |
| Total number of parameters | 135 |
| Total number of training data | 132 |
| Learning algorithm | Gradient back propagation and least squares method |
| Number of fuzzy rulers | 27 |

**Table 2. Final parameters of the membership functions of the piezometer I1.**

| final function | weak | | | average | | | high | | |
|---|---|---|---|---|---|---|---|---|---|
| | ai | bi | Ci | ai | bi | ci | ai | bi | ci |
| level | 1.0947 | 2.3585 | -4.2974 | 0.4738 | 2.5804 | -1.3234 | 1.8114 | 1.6035 | 0.9052 |
| season | 1.5757 | 1.182 | -0.2852 | 1.1378 | 3.3772 | 2.2213 | 1.6374 | 3.3772 | 2.2213 |
| time | 2.7784 | 2.3521 | 0.2099 | 1.5173 | 3.8164 | 5.2822 | 3.8372 | 3.3488 | 8.9724 |

the correlation in the case of the ANFIS model is good and centred around a straight line, while in the HST the values are quite scattered (Fig 8). Table 4 presents the criteria for comparing the different models. We can be see that the assessment of the phenomenon based on the fuzzy neuron model describing the piezometric I1 is good because the mean squared error (MSE) is low and the correlation coefficient high compared to the HST model.

This adequacy is justified by the fact that the model makes it possible to highlight the cross-relationships existing between the different explanatory variables. A gap in the conventional model (HST) which superimposes the effect of these variable is thus corrected. The MAPE criterion (Mean Absolute Percentage Error) is low, which proves that the system does not

**Table 3. Coefficient of final parameters.**

| function | Parameters | | | | |
|---|---|---|---|---|---|
| | Coefficient of the variable level retained: n | of the variable Season: s | of the variable Time: t | b | $w_i$ |
| $out1\_f_1$ | -66.86201 | 14.66457 | 103.99512 | 28.52094 | 0.000718018 |
| $out1\_f_2$ | -13.67075 | 80.16198 | 118.95478 | 32.38700 | 0.002916573 |
| $out1\_f_3$ | 26.12609 | 52.01292 | 37.84743 | 10.14450 | 0.003682125 |
| $out1\_f_4$ | -60.08609 | 34.65492 | 62.39650 | 16.05303 | 0.000398601 |
| $out1\_f_5$ | -21.82759 | 32.43701 | 58.24856 | 16.96844 | 0.001619106 |
| $out1\_f_6$ | 12.12196 | 40.67423 | 55.84260 | 10.86692 | 0.002044095 |
| $out1\_f_7$ | -1.22492 | -20.75657 | -8.41647 | -0.06685 | 0.000799131 |
| $out1\_f_8$ | -8.96193 | 4.18536 | 2.33630 | 9.01214 | 0.003246051 |
| $out1\_f_9$ | -12.70787 | -0.77933 | 43.79042 | 7.22182 | 0.004098085 |
| $out1\_f_{10}$ | -78.95026 | 181.33469 | 71.64195 | 215.4254 | 0.000748617 |
| $out1\_f_{11}$ | 3.25957 | -29.03033 | 125.71012 | 15.11556 | 0.003040863 |
| $out1\_f_{12}$ | 13.82312 | 55.21798 | 37.42386 | -19.59825 | 0.00383904 |
| $out1\_f_{13}$ | 2.05086 | -124.93501 | 191.85175 | 99.11219 | 0.000415587 |
| $out1\_f_{14}$ | 34.83082 | 72.43588 | 89.51021 | 88.88520 | 0.001688105 |
| $out1\_f_{15}$ | -77.30770 | -142.24788 | 55.84266 | 16.87389 | 0.002131204 |
| $out1\_f_{16}$ | -12.24695 | 50.66410 | 115.60437 | 20.02558 | 0.000833186 |
| $out1\_f_{17}$ | -8.64564 | -18.49168 | 109.95610 | 72.26042 | 0.003384382 |
| $out1\_f_{18}$ | 46.76177 | 18.06850 | 52.46139 | 27.78286 | 0.004272726 |
| $out1\_f_{19}$ | -12.06337 | 3.22211 | 0.89449 | 499.6492 | 0.03531372 |
| $out1\_f_{20}$ | -3.19847 | -4.73258 | -1.87640 | 508.2304 | 0.14344351 |
| $out1\_f_{21}$ | 12.88944 | 13.54366 | -1.47723 | 497.2053 | 0.181095058 |
| $out1\_f_{22}$ | -1.21297 | -0.77610 | -0.44070 | 502.0313 | 0.019604057 |
| $out1\_f_{23}$ | 2.44147 | 0.11472 | 0.72269 | 496.5514 | 0.079631225 |
| $out1\_f_{24}$ | -6.96780 | -0.78891 | 0.40666 | 494.3632 | 0.100533104 |
| $out1\_f_{25}$ | 0.56275 | 0.09870 | -0.17007 | 497.8733 | 0.039303016 |
| $out1\_f_{26}$ | -0.85591 | -0.28260 | 0.00612 | 502.0062 | 0.159647936 |
| $out1\_f_{27}$ | 0.04226 | 0.58579 | -0.05504 | 492.7806 | 0.201552878 |

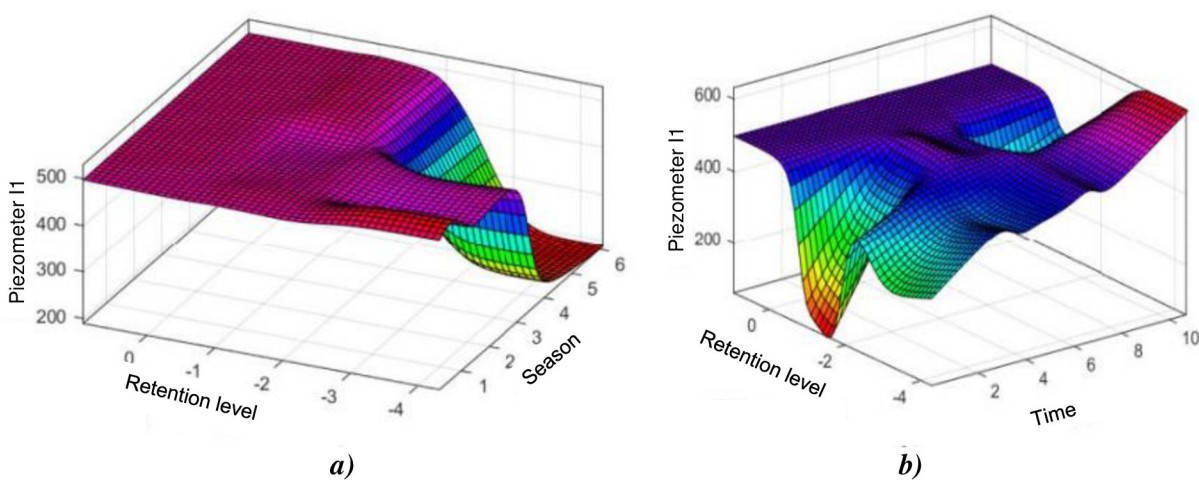

**Fig 6.** Evolution of the level of piezometer I1: a) depending on the water level and season; b) depending on the water level and the weather (lifetime of structure).

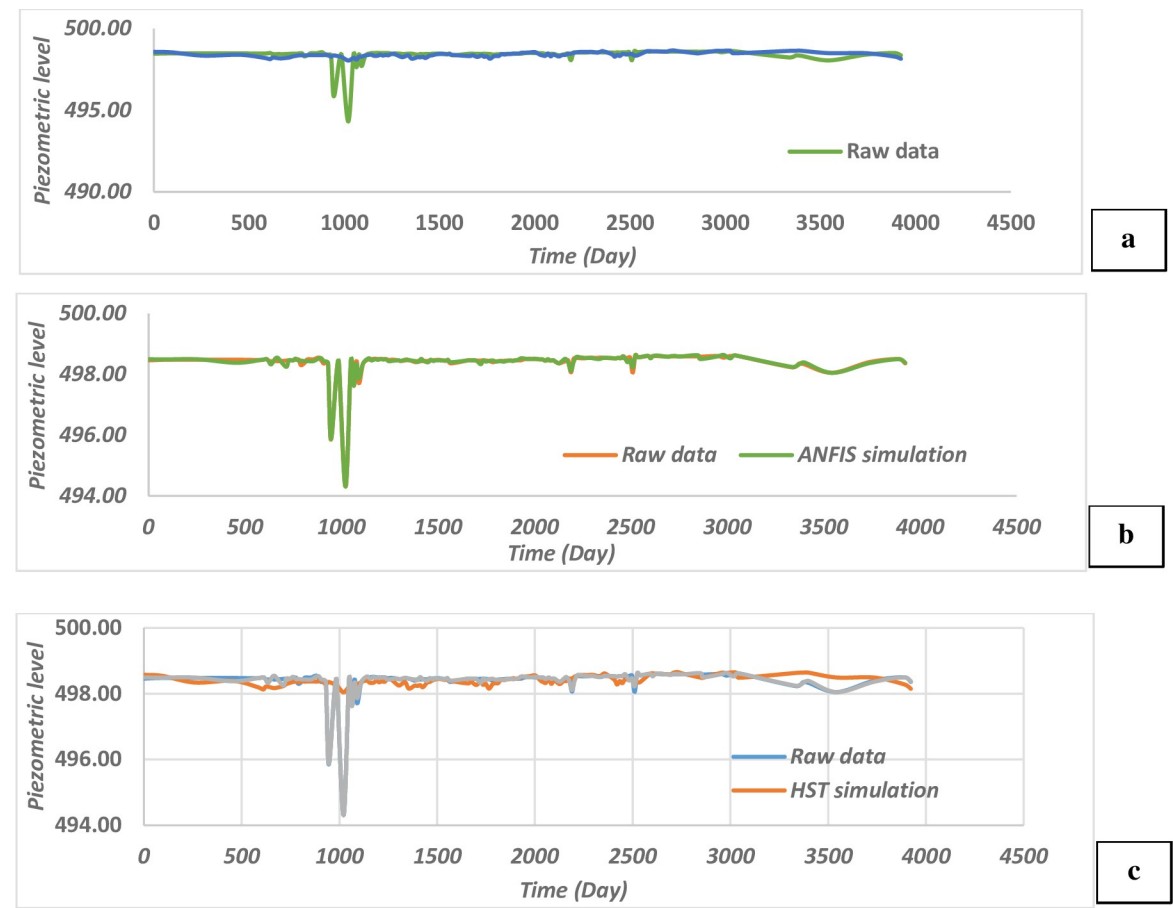

**Fig 7. Comparison of the different behaviour models of the piezometer I1.**

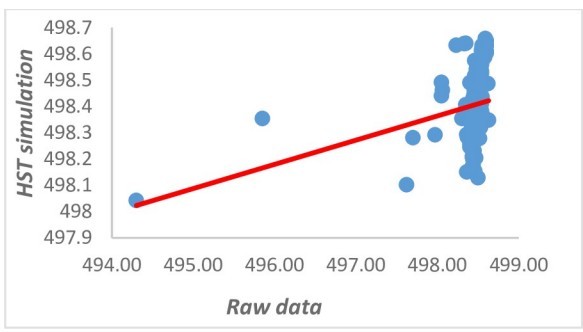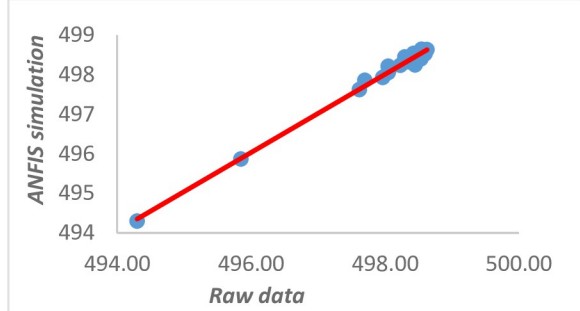

**Fig 8. Comparison between the real data and the simulated data of the piezometer I1.**

correctly predict the behaviour of the piezometer I1 in the short term. D and Dα are parameters of the Kolmogorov-Smirov test.

Assumptions of the Kolmogorov-Smirov test:

- H0: the simulated values are not significantly different from the real values,

- H1: the simulated values are significantly different from the real values.

The principle of the test consists in calculating the cumulative distribution of the theoretical proportions and comparing it with those observed. The test statistic is considered to be: D, the maximum difference in absolute value between the observed cumulative proportions and the simulated cumulative proportions [46–48].

$$D = max\|PcO - PcT\|$$

For the piezometer I1 (ANFIS model),

$$D = 0.00018488$$

The critical value Dα at threshold α = 5% for a sample of size n (n > 35) is given by: $D_{\alpha} = \frac{1.36}{\sqrt{n}}$ either $D_{\alpha} = 0.11837281$

When D < Dα so the hypothesis H0 is accepted. Then the simulated values of the ANFIS model are not significantly different from the real values. The same analysis process is carried out on all the piezometers. The summary in Table 5 presents the different criteria for assessing the model as well as the graphic representations of the piezometers used. In order to well appreciate the efficiency of the fuzzy neuron model, a comparison is made with the HST model according to the data from the various piezometers of the dam. Upon analysis of the correlation indicators presented in Table 5, it clearly appears that the ANFIS model correctly

**Table 4. Comparison of HST and ANFIS.**

|  | Model | HST | ANFIS |
|---|---|---|---|
|  | MSE | 0.18097573 | 0.00249069 |
|  | NASH | -8.8586626 | 0.98734384 |
|  | R | 9.21% | 98.75% |
|  | MAPE | 0.04491726 | 0.00829865 |
| Kolmogorov-Smirnov | D | 0.00015226 | 0.00018488 |
|  | Dα | 0.11837281 | 0.11837281 |

**Table 5. Performance evaluation of the different prediction models on all piezometers (*D*: The maximum difference in absolute value between the observed cumulative proportions and the simulated cumulative proportions; *Dα*: The critical value).**

| Piezometer | model | | HST | ANFIS |
|---|---|---|---|---|
| E1-1 | *MSE* | | *1.01654506* | *0.0391054* |
| | *NASH* | | *0.00876777* | *0.98033811* |
| | *R* | | *50.22%* | *98.09%* |
| | *MAPE* | | 0.21546735 | 0.03328576 |
| | Kolmogorov-Smirnov | *D* | 0.00022238 | 0.00036481 |
| | | *Dα* | 0.11837281 | 0.11837281 |
| I1 | *MSE* | | *0.18097573* | *0.00249069* |
| | *NASH* | | *-8.8586626* | *0.98734384* |
| | *R* | | *9.21%* | *98.75%* |
| | *MAPE* | | 0.04491726 | 0.00829865 |
| | Kolmogorov-Smirnov | *D* | 0.00015226 | 0.00018488 |
| | | *Dα* | 0.11837281 | 0.11837281 |
| I2 | *MSE* | | *0.7199511* | *0.00722191* |
| | *NASH* | | *-1.9900762* | *0.99242235* |
| | *R* | | *25.06%* | *99.25%* |
| | *MAPE* | | 0.1877615 | 0.01529493 |
| | Kolmogorov-Smirnov | *D* | 1.60E-04 | 3.39E-04 |
| | | *Dα* | 0.11837281 | 0.11837281 |
| I3 | *MSE* | | *0.08864675* | *0.00649858* |
| | *NASH* | | *-5.6616545* | *0.93187339* |
| | *R* | | *13.05%* | *93.63%* |
| | *MAPE* | | 0.03171359 | 0.00885452 |
| | Kolmogorov-Smirnov | *D* | 0.00013798 | 0.00015548 |
| | | *Dα* | 0.11837281 | 0.11837281 |
| I8-2L | *MSE* | | *0.00873718* | *0.0031229* |
| | *NASH* | | *-0.1914663* | *0.7567704* |
| | *R* | | *45.63%* | *80.57%* |
| | *MAPE* | | 0.01801691 | 0.0087981 |
| | Kolmogorov-Smirnov | *D* | 1.35E-04 | 1.34E-04 |
| | | *Dα* | 0.11837281 | 0.1183728 |
| M1-1 | *MSE* | | *0.26905319* | *0.0022615* |
| | *NASH* | | *-8.8586626* | *0.9873438* |
| | *R* | | *8.87%* | *99.23%* |
| | *MAPE* | | 0.05758224 | 0.0081616 |
| | Kolmogorov-Smirnov | *D* | 0.00013036 | 0.0001492 |
| | | *Dα* | 0.11837281 | 0.1183728 |
| I4 | *MSE* | | *0.06852541* | *0.00252257* |
| | *NASH* | | *-3.2512176* | *0.96926997* |
| | *R* | | *19.04%* | *97.02%* |
| | *MAPE* | | 0.03758204 | 0.00821245 |
| | Kolmogorov-Smirnov | *D* | 5.47E-05 | 6.18E-05 |
| | | *Dα* | 0.11837281 | 0.11837281 |
| I5-2L | *MSE* | | *6.9733032* | *0.00297777* |
| | *NASH* | | *-8.9315753* | *0.99961191* |
| | *R* | | *9.15%* | *99.96%* |
| | *MAPE* | | 0.31130267 | 0.01013079 |
| | Kolmogorov-Smirnov | *D* | 0.00048475 | 0.00066052 |
| | | *Dα* | 0.11837281 | 0.11837281 |

(*Continued*)

**Table 5.** (Continued)

| Piezometer | model | | HST | ANFIS |
|---|---|---|---|---|
| I6-2L | MSE | | 0.01470121 | 0.00230562 |
| | NASH | | -5.6992833 | 0.84326898 |
| | R | | 12.99% | 86.35% |
| | MAPE | | 0.02440705 | 0.00847497 |
| | Kolmogorov-Smirnov | D | 7.44E-05 | 9.49E-05 |
| | | Dα | 0.11837281 | 0.11837281 |
| I7-2L | MSE | | 0.030974 | 0.00658386 |
| | NASH | | -5.6992833 | 0.84326898 |
| | R | | 27.58% | 84.61% |
| | MAPE | | -1.62570427 | 0.81736629 |
| | Kolmogorov-Smirnov | D | 1.44E-04 | 1.55E-04 |
| | | Dα | 0.11837281 | 0.11837281 |
| M6-2 | MSE | | 0.04371014 | 0.00889755 |
| | NASH | | -7.0579013 | 0.78665187 |
| | R | | 11.04% | 81.89% |
| | MAPE | | 0.02065605 | 0.01549152 |
| | Kolmogorov-Smirnov | D | 1.27E-04 | 1.39E-04 |
| | | Dα | 0.11837281 | 0.11837281 |
| M7-2 | MSE | | 0.01880582 | 0.00463144 |
| | NASH | | -0.595434 | 0.82019766 |
| | R | | 38.53% | 84.86% |
| | MAPE | | 0.02345939 | 0.0096185 |
| | Kolmogorov-Smirnov | D | 1.56E-04 | 1.52E-04 |
| | | Dα | 0.11837281 | 0.11837281 |
| M2-1 | MSE | | 0.7199511 | 0.00722191 |
| | NASH | | -1.9900762 | 0.99242235 |
| | R | | 25.06% | 99.25% |
| | MAPE | | 0.1877615 | 0.01529493 |
| | Kolmogorov-Smirnov | D | 1.60E-04 | 3.39E-04 |
| | | Dα | 0.11837281 | 0.11837281 |
| M3-1 | MSE | | 0.2891059 | 0.0034373 |
| | NASH | | -3.2128611 | 0.99029053 |
| | R | | 19.18% | 99.04% |
| | MAPE | | 0.08921857 | 0.0097318 |
| | Kolmogorov-Smirnov | D | 1.13E-04 | 1.60E-04 |
| | | Dα | 0.11837281 | 0.11837281 |
| M4-1 | MSE | | 0.02583915 | 0.00258702 |
| | NASH | | -0.4467653 | 0.9368043 |
| | R | | 40.87% | 94.08% |
| | MAPE | | 0.0346867 | 0.00819494 |
| | Kolmogorov-Smirnov | D | 8.47E-05 | 8.69E-05 |
| | | Dα | 0.11837281 | 0.11837281 |
| M5-1 | MSE | | 0.05308272 | 0.00940327 |
| | NASH | | 0.5238724 | 0.93941527 |
| | R | | 67.74% | 94.29% |
| | MAPE | | 0.0367271 | 0.01289637 |
| | Kolmogorov-Smirnov | D | 0.00032319 | 0.00030788 |
| | | Dα | 0.11837281 | 0.11837281 |

(*Continued*)

**Table 5.** (Continued)

| Piezometer | model | | HST | ANFIS |
|---|---|---|---|---|
| M5-2 | MSE | | 0.03553419 | 0.00063743 |
| | NASH | | -0.4264033 | 0.98932728 |
| | R | | 41.21% | 98.95% |
| | MAPE | | 0.21546735 | 0.03328576 |
| | Kolmogorov-Smirnov | D | 0.00022238 | 0.00036481 |
| | | Dα | 0.11837281 | 0.11837281 |
| M8-2 | MSE | | 0.01880582 | 0.00463144 |
| | NASH | | -0.595434 | 0.82019766 |
| | R | | 38.53% | 84.86% |
| | MAPE | | 0.02345939 | 0.0096185 |
| | Kolmogorov-Smirnov | D | 1.56E-04 | 1.52E-04 |
| | | Dα | 0.11837281 | 0.11837281 |
| M8-1 | MSE | | 0.00460024 | 0.01644953 |
| | NASH | | -0.118744 | 0.82498519 |
| | R | | 47.20% | 85.23% |
| | MAPE | | 0.02308931 | 0.01074947 |
| | Kolmogorov-Smirnov | D | 1.48E-04 | 1.34E-04 |
| | | Dα | 0.11837281 | 0.11837281 |
| M9-1 | MSE | | 0.97957829 | 0.17354932 |
| | NASH | | -0.0800774 | 0.89717252 |
| | R | | 48.08% | 90.80% |
| | MAPE | | 0.19712565 | 0.0621059 |
| | Kolmogorov-Smirnov | D | 1.67E-04 | 2.22E-04 |
| | | Dα | 0.11837281 | 0.11837281 |
| M10-1 | MSE | | 1.60972866 | 0.05971698 |
| | NASH | | 0.00337126 | 0.9809826 |
| | R | | 50.08% | 98.15% |
| | MAPE | | 0.26691034 | 0.04057676 |
| | Kolmogorov-Smirnov | D | 3.26E-04 | 4.17E-04 |
| | | Dα | 0.11837281 | 0.11837281 |
| M10-2 | MSE | | 0.88717832 | 0.03048639 |
| | NASH | | -0.0773726 | 0.98172326 |
| | R | | 48.14% | 98.22% |
| | MAPE | | 0.19400747 | 0.02962152 |
| | Kolmogorov-Smirnov | D | 2.83E-04 | 4.20E-04 |
| | | Dα | 0.11837281 | 0.11837281 |

predicts the phenomenon investigated with correlation coefficient R greater than 80% on the data from all the piezometers.

**4.1.2. Piezometer M1-1.** The basic model (HST) tends to generalize the system and approximates the behaviour of the piezometer more or less well until 2015 when the peaks become larger and larger. From then on the HST model does not can't keep up anymore. An assumption of the HST method is that «errors are distributed according to the normal distribution $\varepsilon\_i \equiv N(0, \sigma\_\varepsilon)$» but in the case of the piezometer M1-1, from 2015, the empirical distribution of residues no longer fits with the Normal distribution. Fig 9B shows the effect of water level and time on the level of piezometer M1-1. It will be notice that the water level in

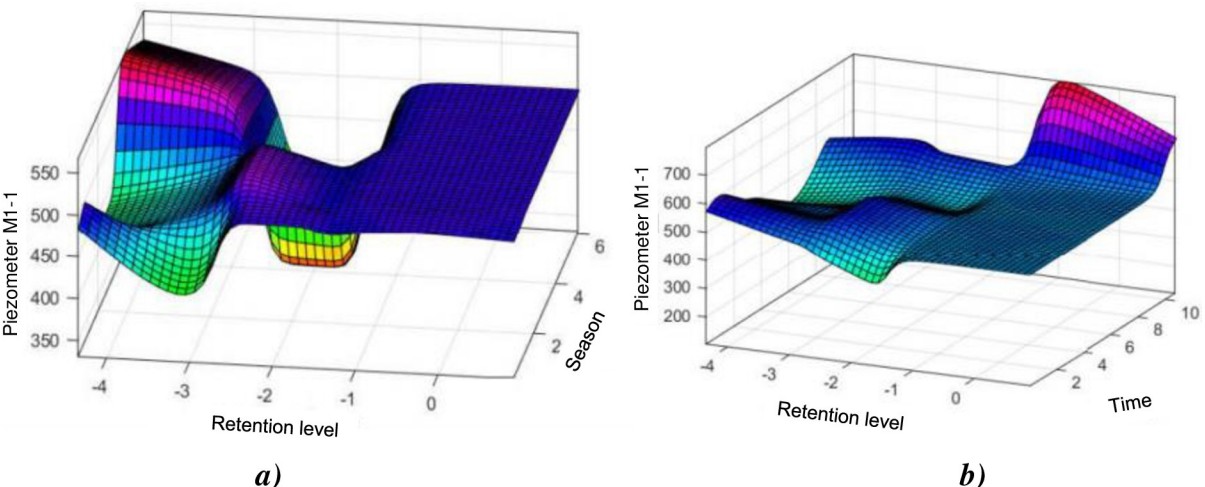

**Fig 9.** Evolution of the piezometer level: a) M1–1 as a function of the water level and the season; b) M1–1 depending on the water level and time (lifetime of the structure).

the dam significantly influences the piezometry. It is noted in Fig 9A, that the season acts in a sinusoidal way on the coast of the piezometer M1-1. It is also interesting to note that in neighbourhoods of the NR (Normal Reservoir), the piezometric level does not change almost whatever the season, this shows the preponderance of the influence once again of the level of the reservoir with respect to the season.

**4.1.3. Piezometer I2.**    Fig 10A shows us the influence of the level of the reservoir and time on the level of the piezometer I2, we notice immediately that as for the previous piezometers, the level of the reservoir has a significant influence on the piezometric level, in fact, we notice that around 0 the piezometric level hardly changes regardless of the time, yet logic would dictate that the consolidation of the ground over time also leads to a decrease in the influence of the level over time.

Fig 10B shows us the influence of the level of the reservoir and the season on the level of the piezometer I2, we notice immediately that as for the previous piezometers, that the season remains mute compared to the level of the reservoir at the around the normal retention level,

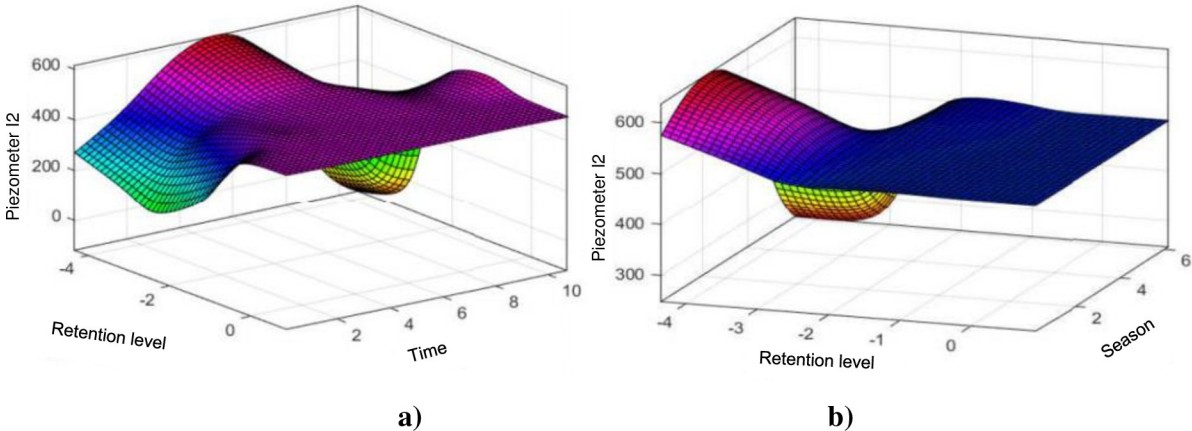

**Fig 10.** Evolution of the piezometer I2 level: a) Depending on the water level and time; b) According to the coast of the water and the season.

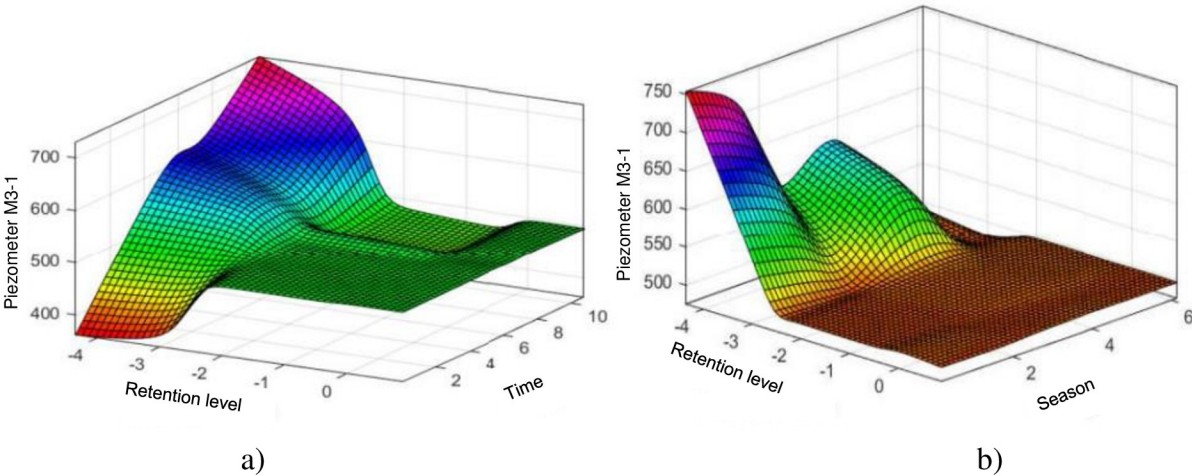

**Fig 11.** Evolution of the elevation of the M3–1 piezometer: a) Depending on the water level and time (lifetime of the structure); b) According to the coast of the water and the season.

thus the level of the reservoir once again demonstrates the preponderance of its influence whatever the season, even ignoring the contraction and expansion of the concrete.

**4.1.4. M3-1 piezometer.** The ANFIS model is more compatible with the data observed on the M3-1 piezometer, indeed it manages to predict all the variations and follows the peaks very well unlike the HST model which has great difficulty in predicting the behaviour of the piezometer at the level of sudden variations. One notices again the domination of the influence of the piezometry in the neighbourhoods of the normal reservoir, the weather hardly influences the level of the piezometry when the level of the reservoir is normal, which is almost always the case. We can underline the constant character whatever the season of the piezometric level in the vicinity of 0 which represents the average of the levels of reservoir, which shows the strong influence of the level of the reservoir compared to the season on the behaviour of the piezometer M3-1 (Fig 11A and 11B).

**4.1.5. Piezometer I3.** One can immediately appreciate the power of the ANFIS model which apprehends the behaviour of the piezometer very well and which faithfully predicts the results of the piezometer I3 compared to the HST model which always has a lot of difficulty to predict peaks, even if the prediction is acceptable. It is important to emphasize, apart from the strong influence of the reservoir level, that the piezometric levels remain much higher upstream than downstream, which testifies to the effectiveness of the scaling layer and the non-communication of water between the two galleries. The piezometric levels downstream remain lower than the levels upstream regardless of the season, which testifies to the effectiveness of the scaling layer. Note also here the strong influence of the level of the reservoir (Fig 12A and 12B).

**4.1.6. Piezometer M8-1.** Fig 13A and 13B show the evolution of the water level in this piezometer as a function of time. The ANFIS model is more compatible with the data observed on the M8-1 piezometer, indeed it manages to predict all the variations and follows the peaks very well unlike the HST model which great difficulty in predicting the behaviour of the piezometer at the level of sudden variations.

**4.1.7. Piezometer I8-2L.** Fig 14A shows us the influences of the reservoir level and time on the I8-2L piezometer level, as for all the other piezometers of the intake dam, the reservoir level has significant influence on the piezometer level which hardly varies around 0. Fig 14B

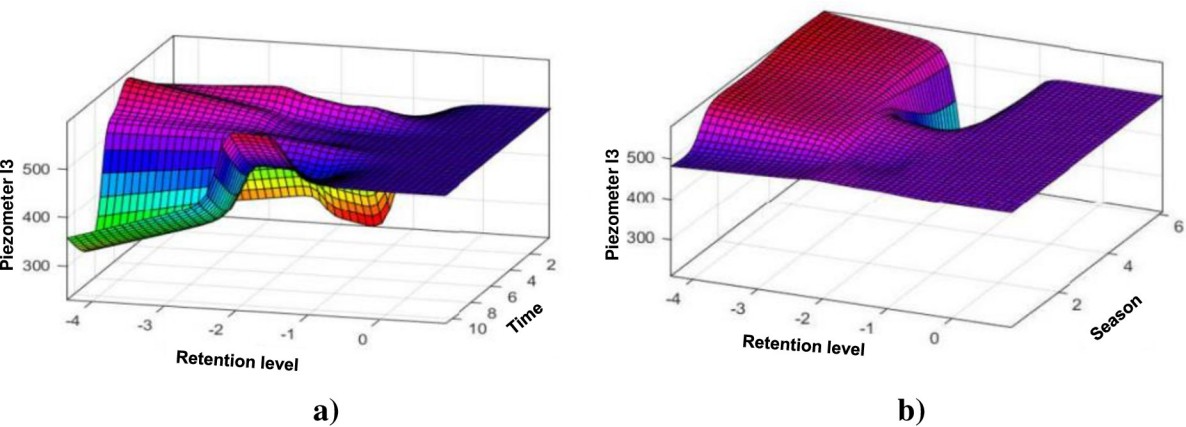

**Fig 12.** Evolution of the elevation of the piezometer I3: a) as a function of the elevation of the water level and time (lifetime of the structure); b) depending to the water coast and the season.

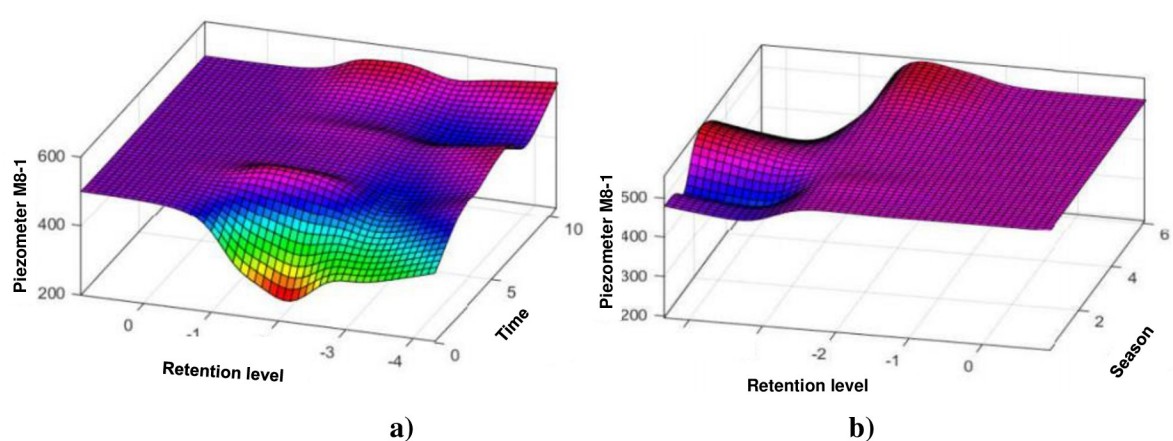

**Fig 13.** Evolution of the elevation of the piezometer M8–1: a) depending on the elevation of the water and time (lifetime of the structure); b) according to the elevation of the water and the season.

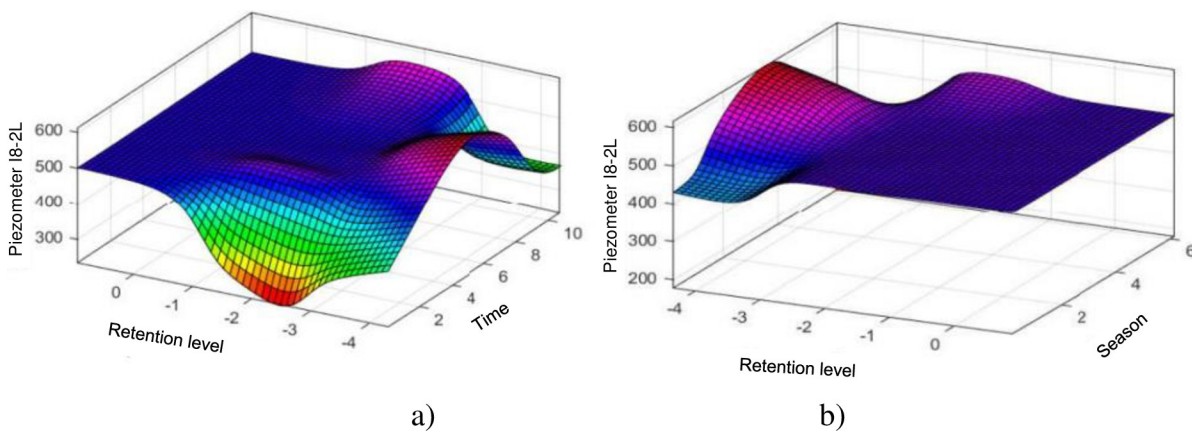

**Fig 14.** Evolution of the elevation of the piezometer I8–2L: a) depending on the elevation of the water and time (lifetime of the structure); b) according to the elevation of the water and the season.

shows the influences of the level of the reservoir and the season on the piezometer level I8-2L, as for all the other piezometer level which hardly varies around 0.

**4.1.8. Piezometer M9-1.** I can be seen that the two models apprehend the behaviour of the M9-1 piezometer quite well when it operates in an almost linear manner (Fig 15C). But the various observable peaks are well estimated by the ANFIS model (Fig 15B) while the basic HST model (Fig 15A) generalizes the behaviour of the piezometry. Fig 16A, shows the influence of the level of the reservoir and of time on the piezometer level M9-1, unlike the other piezometers of the intake dam, the influence of the level of the reservoir drops significantly and a remarkable increase is observed the influence of time, with a piezometric level which increases with time. Fig 16B shows the influence of the reservoir level and the season on the M9-1 piezometer level, unlike the other piezometers of the intake dam, the influence of the reservoir

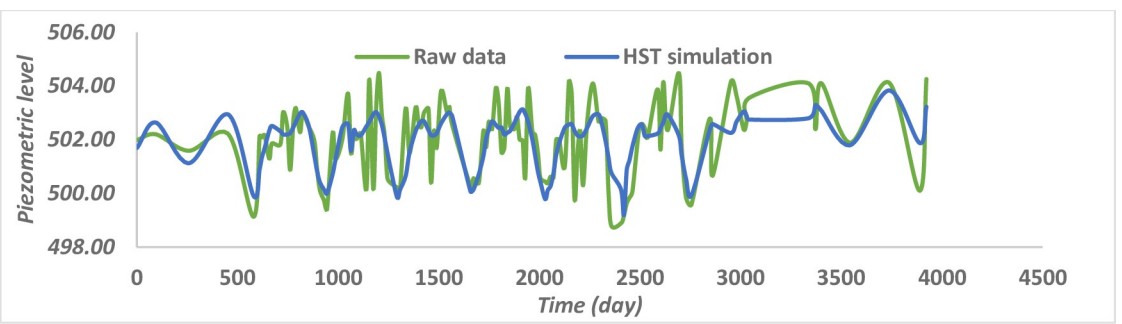

a)

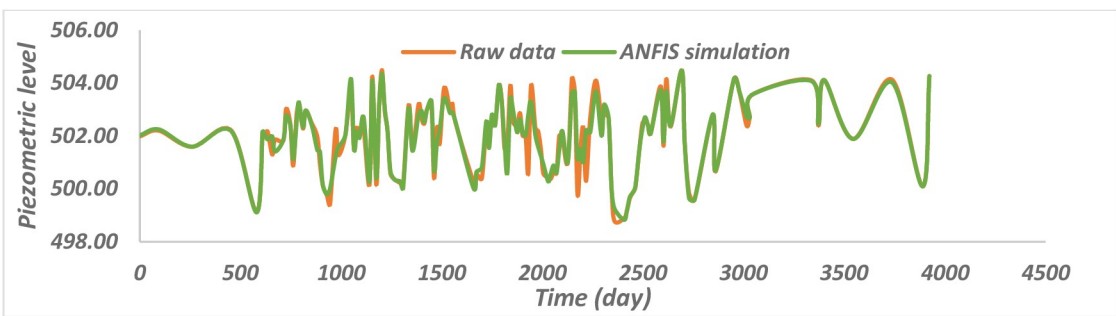

b)

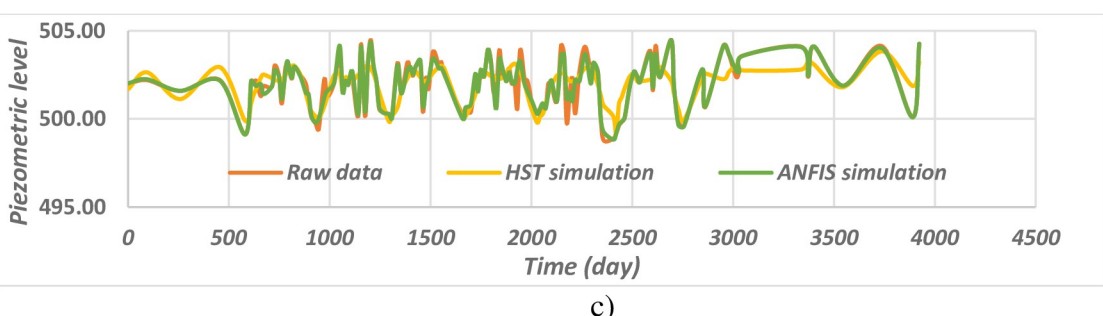

c)

**Fig 15. Comparison of the different behaviour models of the M9–1 piezometer.**

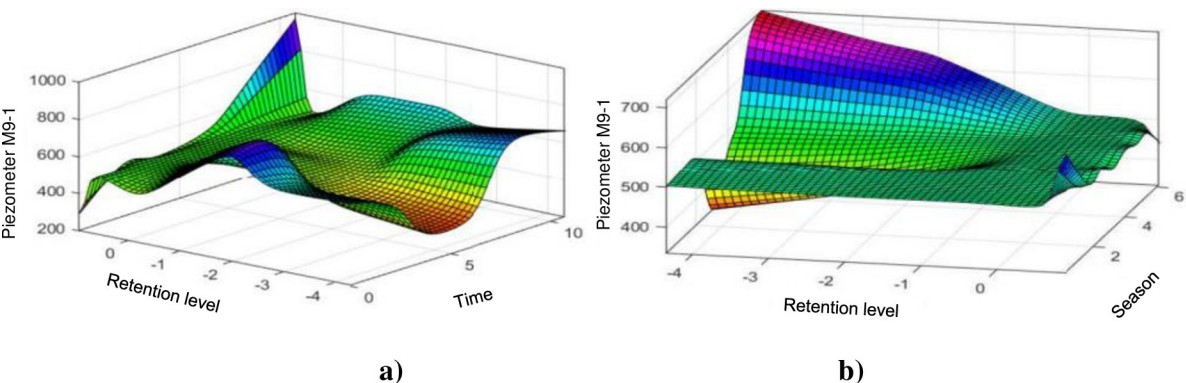

**Fig 16.** Evolution of the elevation of the piezometer M9–1: a) depending on the coast of the water and the weather; b) according to the coast of the water and the season.

level drops significantly and an increase remarkable the influence of the season through slight oscillations around the level of the normal reservoir.

**4.1.9. Piezometer M10-1.** It can be seen that the two models apprehend quite well the behaviour of the M10-1 piezometer when it operates in an almost linear manner. The Fig 17A, highlights the influences of the level of the reservoir and of time on the level piezometer M10-1. The influence of the level of the reservoir decreases significantly and one observes a remarkable increase in the influence of the time better although on the M10-1 piezometer, whit a piezometric level which increases with time, should we conclude that it is linked to the fact we are approaching the spillways dam? Fig 17B shows us the influences of the level of the reservoir and the season on the piezometer level M10-1, the influence of the level of the reservoir drops significantly and we observe a remarkable increase in the influence of the season through slight oscillations around the normal restraint level. The behaviour of the dam through all piezometers is represented in Table 5. This behaviour is typical on the ground of this region [49].

## 4.2. Spillway dam: Piezometric data analysis

Fig 18 shows the influences of the level of the reservoir and of the season on the piezometer level E1-1. The influence of the level of the reservoir decreases significantly and we observe a remarkable increase in the influence of the season through oscillations around the normal restraint level. The results presented in this part for the analysis of the sub pressures show that

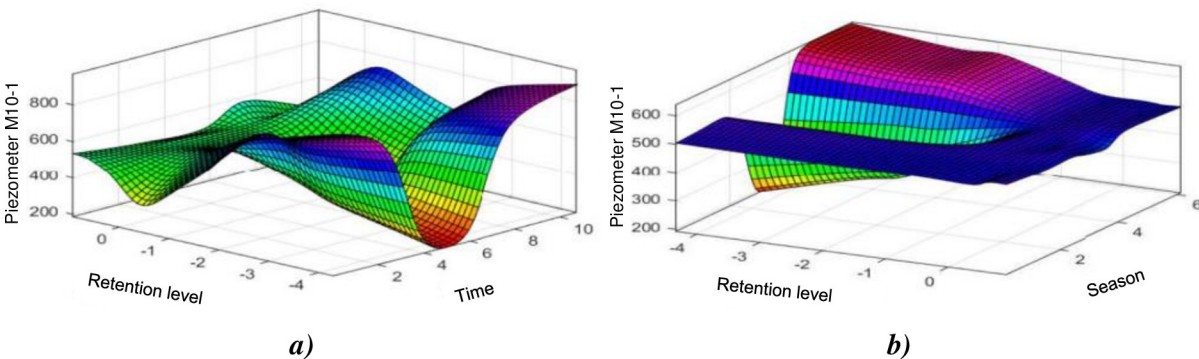

**Fig 17.** Evolution of the M10–1piezometer elevation: a) as a function of the water level and the weather (lifespan of the structure); according to the elevation of the water and the season.

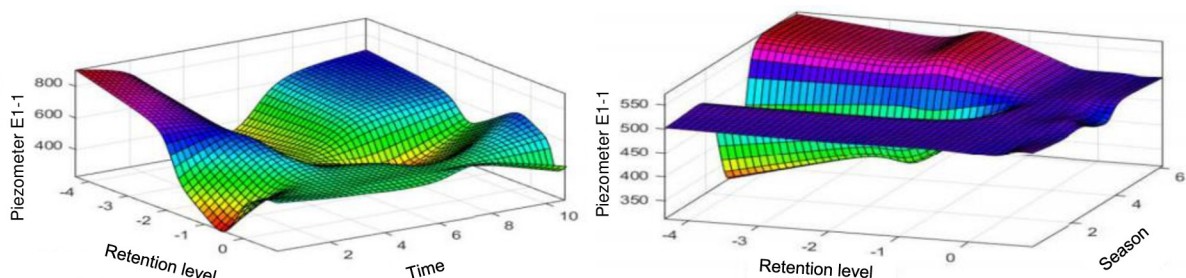

**Fig 18. Evolution of the elevation of the E1–1 piezometer as a function of the water elevation and time and depending on the water elevation and the season.**

the water level in the dam significantly influences the piezometry at the level of the intake dam, we note at the level of this same dam that when we are at the normal reservoir level, the piezometric level hardly changes whatever the season or the whatever, apriority this observation is worrying since the consolidation of the ground should logically reduce the influence of the level of the reservoir over time, but the lack of data from past periods and on the geology of the soil makes any conclusion premature; this beings the case, it is nonetheless certain that the height of the reservoir has a preponderant influence on the piezometric level of the intake dam. From plot 9 (in the vicinity of the spillway dam) there is a significant increase in the influences of the season and the weather and a significant decrease in the influence of the level of the reservoir on the piezometry. This analysis remains valid throughout. Along the spillway dam, these fluctuations can be explained by the variations in activity on this dam according to the seasons (low water level, indeed during these periods, the spillways are open to evacuate the excess water in the dam, and are closed during periods of low flood), this dam is also the most affected by the Alkali-Granulate-Reaction, which helps to observe strong seasonal variations.

## 4.3. Displacement measurements from 1997 to 2019

**4.3.1. Displacements in the pendulum P1D.** After applying the HST and ANFIS models to the raw data of the P1D pendulum, the results presented through Fig 19A and 19B were obtained. Overall, the behaviour of the pendulum is well assimilated by the models, and as shown in the figures above, the errors are acceptable; nevertheless, the excellence of the model is mainly explained by the quality of the influences of the explanatory data which are the level of the reservoir, the season in the year and the time (HST). The statistical analysis of the influential parameters recorded in this pendulum is op good quality (correlation coefficient $R^2$ = 0.99412 for Upstream-Downstream and $R^2$ = 0.97767 for left-right). It shows the low influence of the season on the value of the measurements (4.949% contribution to the explanation of the phenomena for Upstream-Downstream and 17.236% for left-right- «Seasonal effects» graphs). The analysis shows an upward temporal drift, far too upward moreover, of the variations recorded on the pendulum (95.151% explanation for Upstream-Downstream and 92.095% for left-right) resulting in large movements unstabilized 7.44 cm for Upstream-Downstream and 7.11 cm for left-right in 20 years (« Time effects » graphs). A weakly explanatory hydrostatic effect (2.36% for Upstream-Downstream and 2.996% for left-right) resulting in a displacement of 0.39 cm for a rise in the water level of 1 m Upstream-Downstream and by a displacement of 0.1 cm for a rise in the level of the water body of 1 m in left-right bank (see « Hydrostatic Effects » graphs).

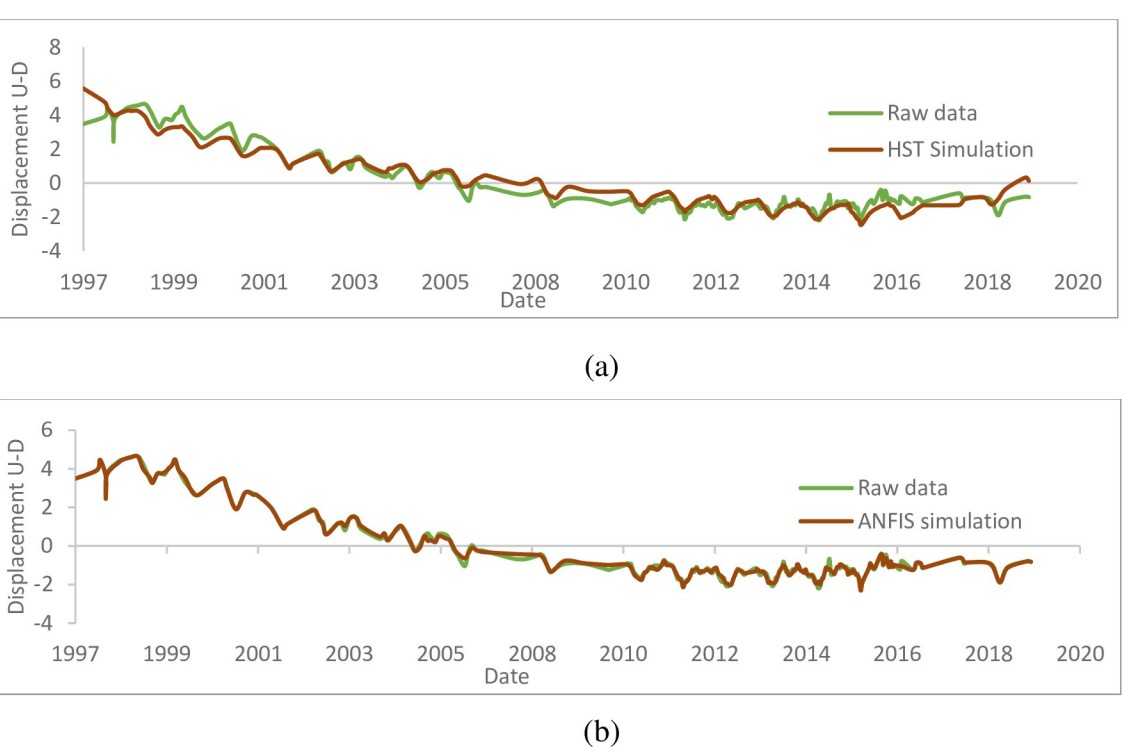

**Fig 19.** Displacements from upstream to downstream measured (cm) on the pendulum P1D; a) HST, b) ANFIS.

**4.3.2. Displacements in the pendulum P3I.** Fig 20A and 20B, highlight the evolution of displacements in the dam according to the years of service. The influence of the explanatory data of all the pendulums is illustrated through Table 6.

### 4.4. Relocation of the spillway dam

**4.4.1. Pendulum E2-3D.** Figs 21 and 22A and 22B show the evolution of the displacements recorded in this pendulum over the years. Table 6 summarize the influences of the reservoir coast, season and time in the E2-3D pendulum.

**4.4.2. Pendulum E6-7D.** Table 6 summarize the influences of the reservoir coast, season and time in the E6-7D pendulum. Fig 23A and 23B show the evolution of displacements as a function of time. The analysis of the monitoring data of the pendulums of the large earthen dam of Songloulou gives the results above. These results are presented in the form of graphs for each of the explanatory effects resulting from the associated statistical analysis. This analysis is good quality. It shows the low (average) influence of the season on the value of the measurements. The analysis also shows an upward temporal drift, far too upward moreover, of the variations recorded on the pendulum resulting in large unstabilized movements and a weakly explanatory hydrostatic effect for explanation. This very strong influence of time, although abnormal, is logically explained by the phenomena of alkali-aggregate-reaction, which promotes accelerated aging of the structure. The presence of many cracks on the structure could also contributes. From the analysis of the data of the various pendulums, it appears that the displacements on the spillway dam are very important.

This phenomenon is explained by the fact that this dam is affected by the alkali-aggregate-reaction and by the fact of significant stresses during periods of flooding (openings and closings of the spillways). In order to obtain the behaviour indicators at any point of this earth

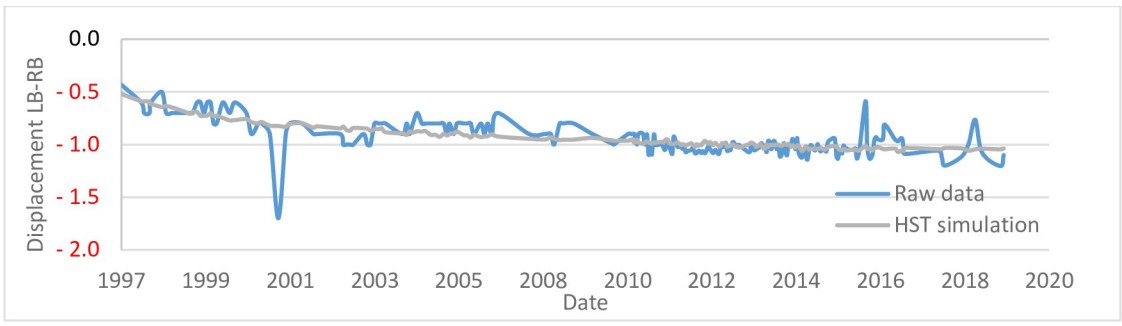

*a)*

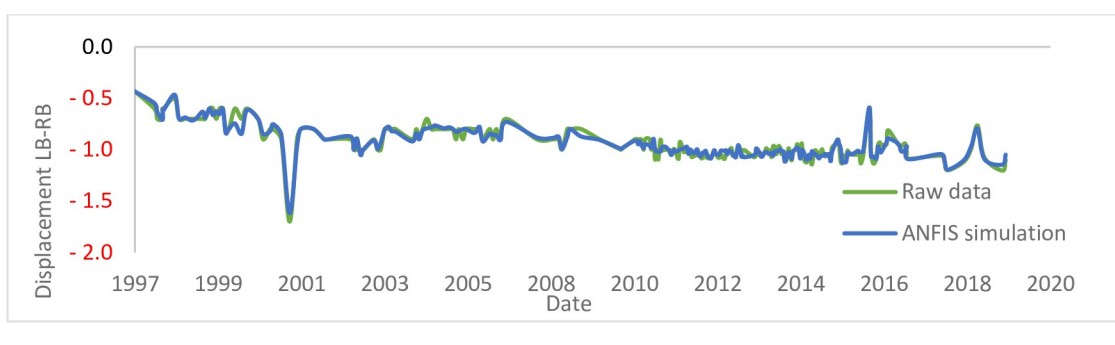

*b)*

**Fig 20.** Displacement (cm) Left–Right bank of the pendulum P3I: a) HST; b) ANFIS.

**Table 6. Evaluation of the performances of the different prediction models on all the pendulums.**

| Pendulum | Model | HST | ANFIS |
|---|---|---|---|
| E6-7D | MSE | 0.26480258 | 1.28E-05 |
| | NASH | 0.81610649 | 0.99999248 |
| | R | 0.84467056 | 0.99999248 |
| | MAPE | -3.72081555 | -0.02277325 |
| E2-3D | MSE | 0.0587672 | 1.47E-06 |
| | NASH | 0.99864049 | 0.99999997 |
| | R | 0.99864234 | 0.99999997 |
| | MAPE | -0.51834962 | -0.00160254 |
| P5I | MSE | 0.03667014 | 1.29E-08 |
| | NASH | 0.92078521 | 0.99999997 |
| | R | 0.9265996 | 0.99999997 |
| | MAPE | 2.16593849 | 0.00104804 |
| P3I | MSE | 0.02367039 | 0.00659559 |
| | NASH | 0.98203608 | 0.99505795 |
| | R | 0.98235309 | 0.99508281 |
| | MAPE | 2.46122631 | 1.20555695 |
| P1D | MSE | 0.41384878 | 0.01361954 |
| | NASH | 0.85262718 | 0.99575404 |
| | R | 0.87155629 | 0.99577299 |
| | MAPE | -62.5843528 | -3.36682009 |

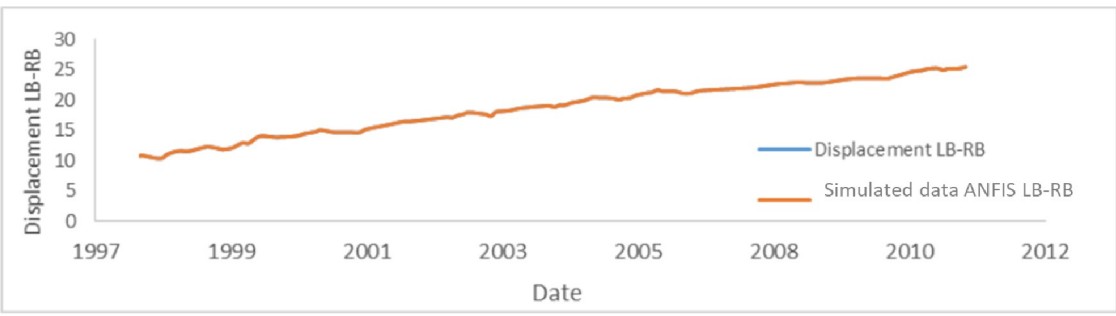

**Fig 21. Displacement (cm) of the Left–Right bank of the pendulum E2–3D.**

dam under different water level, a numerical modelling is carried out from GEOSTUDIO calculation software for analysed the displacements, the stresses and slope stability.

## 5. Numerical simulation of the water level of the earthen dam on GEOSTUDIO

GEOSTUDIO is a Finite element calculation code to perform any type of mechanical analysis for various geotechnical structures [50]. The SIGMA/W module of GEOSTUDIO was used to

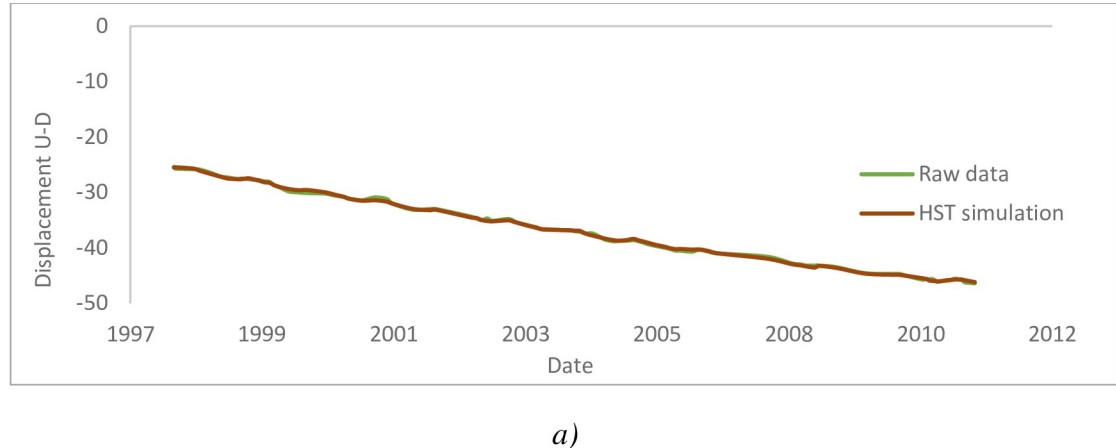

*a)*

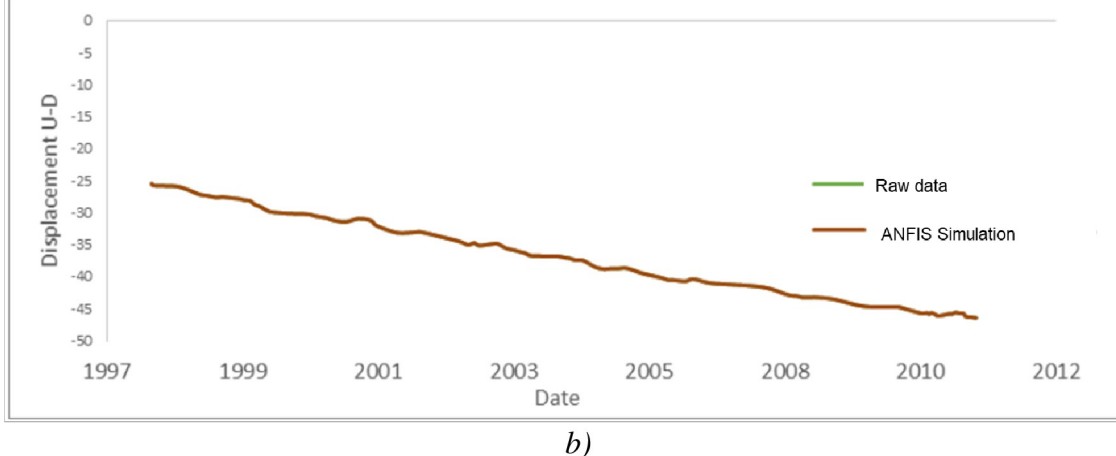

*b)*

**Fig 22.** Displacement (cm) upstream–downstream pendulum E2–3D: a) HST; b) ANFIS.

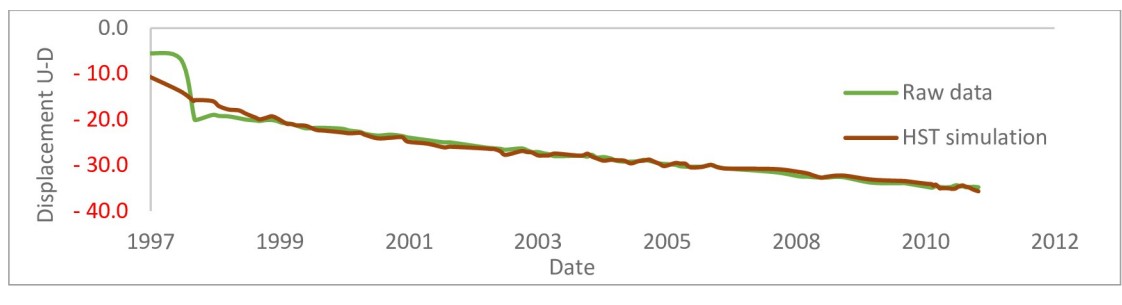

*a)*

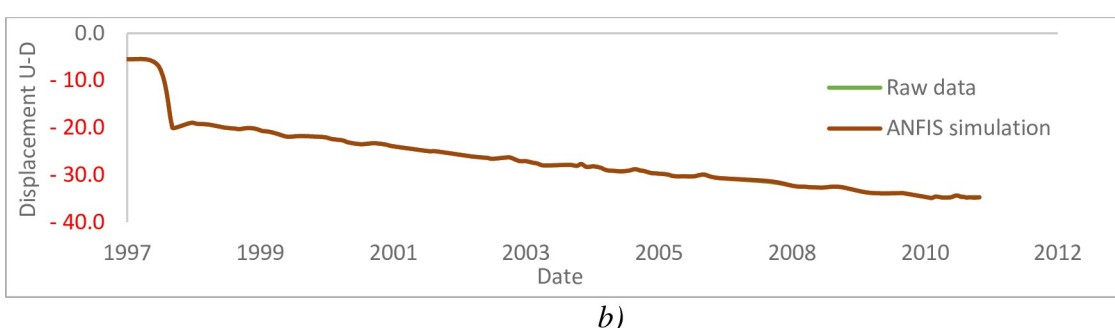

*b)*

**Fig 23.** Upstream–Downstream displacements of pendulum E6–7D: a) HST; b) ANFIS.

obtain the stresses, strains and displacements of the Finite element Model. The water intake dam is modelled in 2D plane strains representing its cross section. The mesh of the model has 6210 triangular elements with 3 nodes. All displacements and seepage or flow condition are fixed at the bottom of the model. The mechanical behaviour of the dam materials obeys of the Mohr-Coulomb failure criterion [51, 52]. Table 7 presents the parameters of the different materials of the water intake dam. In order to ensure the reliability of the results, the meshing of the model was carried out according to the Mestat recommendations for earth dams [53, 54].

The SLOPE module of GEOSTUDIO was used to assess the slope stability of the earth dam. This module uses the method of resolution by limit equilibrium. Bishop's method was used. From the water levels in the reservoir, observed according to the seasons, three piezometric levels were selected for the modelling (minimum: 522 m, normal: 528 m and maximum level: 530 m). The model is subject to loading relative to the weight of the materials and the water pressure from the front side. In order to appreciate the overall behaviour of the structure, the results in stresses, displacements and factor of safety of slope are given in cartographic form on Figs 24–39 and Table 8.

**Table 7. Parameters of subsoil and earth materials.**

| Material | Unit weight of soil: $\gamma_{unsat}$ (kN/m$^3$) | Saturated unit weight of soil: $\gamma_{sat}$ (kN/m$^3$) | Cohesion in terms of effective stress: c' (kPa) | Effective angle of shearing resistance: $\Phi'$ (°) | Young's modulus (MPa) | Poisson's ratio (-) | Coefficient of permeability (m/s) |
|---|---|---|---|---|---|---|---|
| Core | 18 | 19 | 15 | 32 | 74 | 0.3 | 1.15741E-14 |
| Rip-Rap | 22 | 24 | 0 | 40 | 80 | 0.3 | 1.15741E-12 |
| Filtered | 19 | 22 | 0 | 40 | 400 | 0.3 | 2.3148E-12 |
| Rockfill | 22 | 24 | 0 | 45 | 800 | 0.3 | 1.15741E-12 |
| Rock foundation | 26 | 26 | 1000 | 45 | 5000 | 0.3 | 5.78704E-14 |

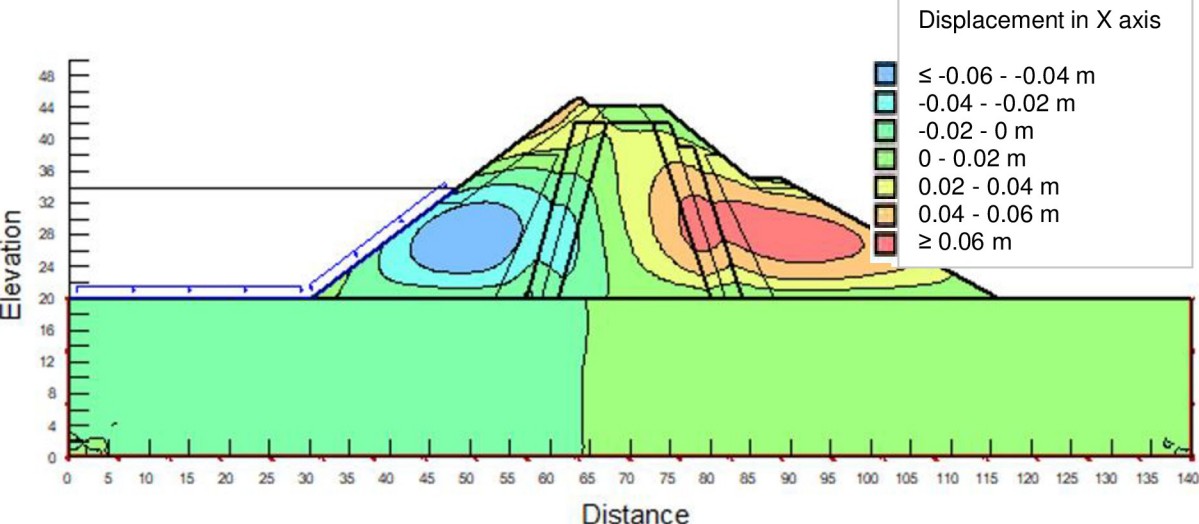

**Fig 24. Horizontal displacements at the water level 522 m.**

In order to validate our GEOSTUDIO calculation program, the results of this part were compared with those of other software commonly used in research and industrial environments. Two other softwares were used. These results concern horizontal displacements with the Finite element code Cast3M [55] and Finite element code Plaxis [56] at retention level 528 and 530 m of the dam. The horizontal displacements calculated are: 11.68 cm; 12 cm; 11.90 cm (at retention level 528 m) and 20 cm; 19cm and 19.47 cm (at retention level 530 m) respectively for GEOSTUDIO, Cast3M and Plaxis softwares (see Figs 28 to 30 and 34 to 36).

At the minimum water level 522 m, the Figs 24 to 27 show the actual behaviour of the earthen dam. The greatest displacements are obtained at the top of the structure. This shear

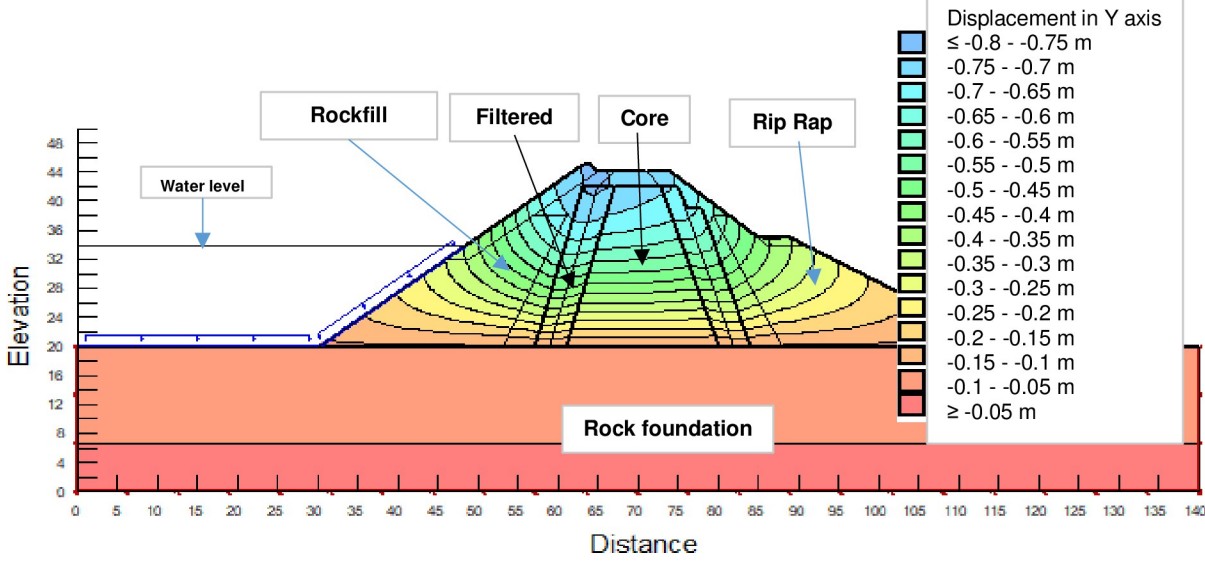

**Fig 25. Vertical displacements at the water level 522 m.**

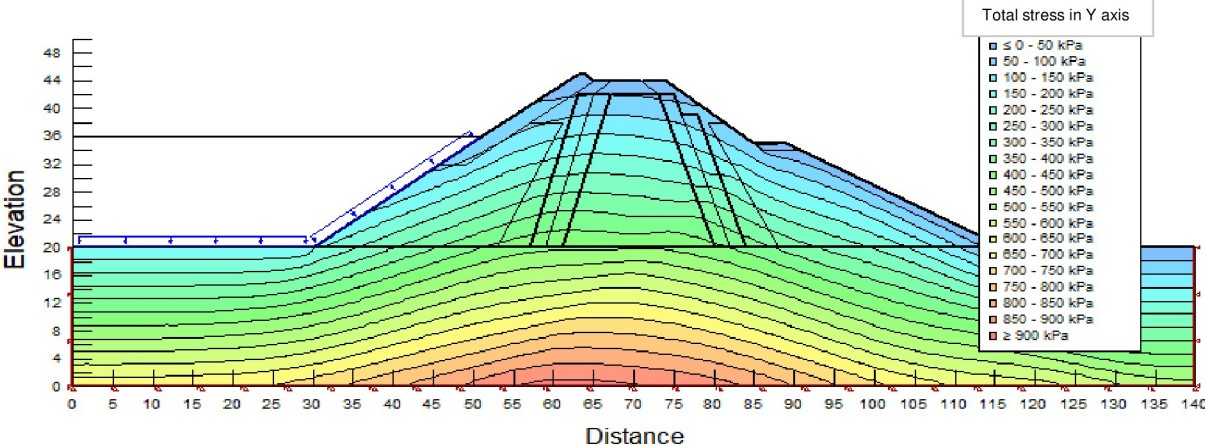

**Fig 26. Total stress at the water level 522 m ($\geq$ 900 kPa at the bottom of the earth dam).**

failure curve of the dam is clearly visible, reflecting the most vulnerable part of the dam subjected to fluctuations in the water level.

The maximum horizontal displacement is 6 cm and vertical displacement is 74 cm. The maximum total stress at the minimal water level is located at the bottom of the model, and is 900 kPa.

Monitoring data did not concern the measurements of vertical displacements in the dam. These vertical displacements in the model are given as a complement to know the global behaviour of the dam. Only the calculated horizontal displacements are compared to the results of the pendulums.

At the normal water level 528 m, the Figs 28 to 33 show the actual behaviour of the earthen dam. The greatest displacements are obtained at the top of the earth dam. The shear failure line of the dam is clearly visible, reflecting the most vulnerable part of the dam subjected to fluctuations in the water level. The maximum horizontal displacement is 11.68 cm and vertical displacement is 80 cm.

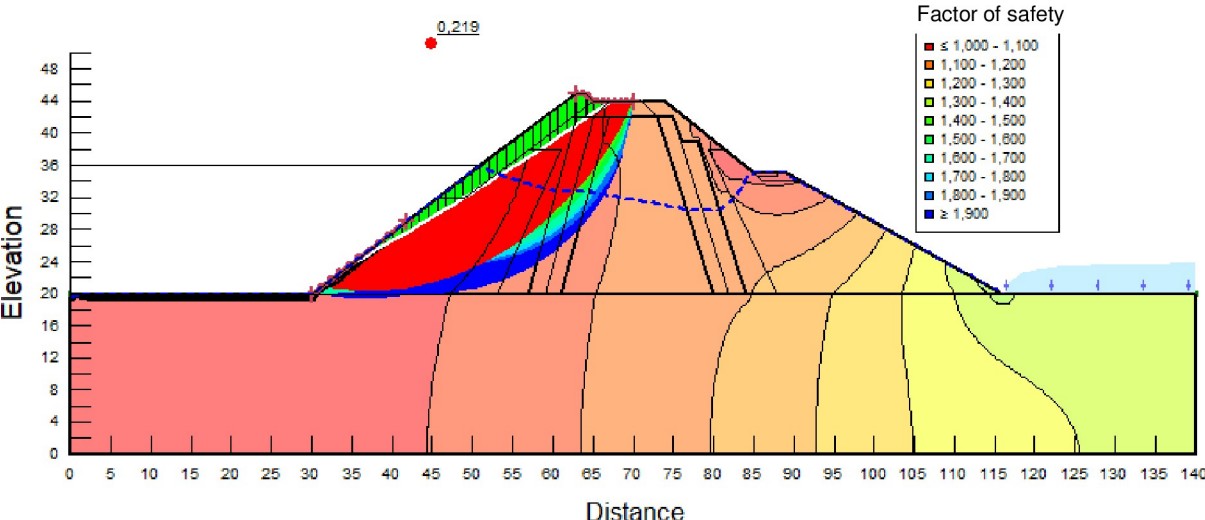

**Fig 27. Critical slope stability of earthen dam of Songloulou at the water level 522 m.**

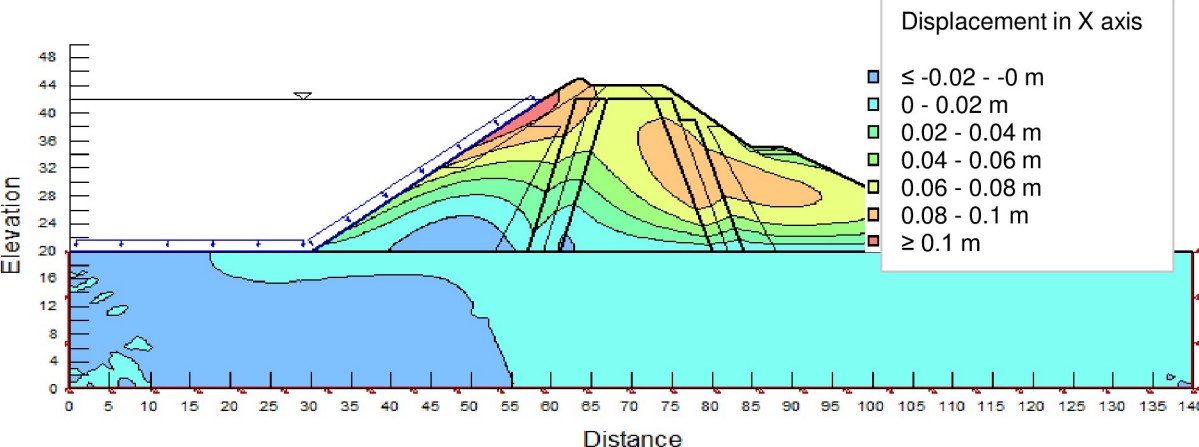

**Fig 28. Horizontal displacements at the water level 528 m obtained by GEOSTUDIO software.**

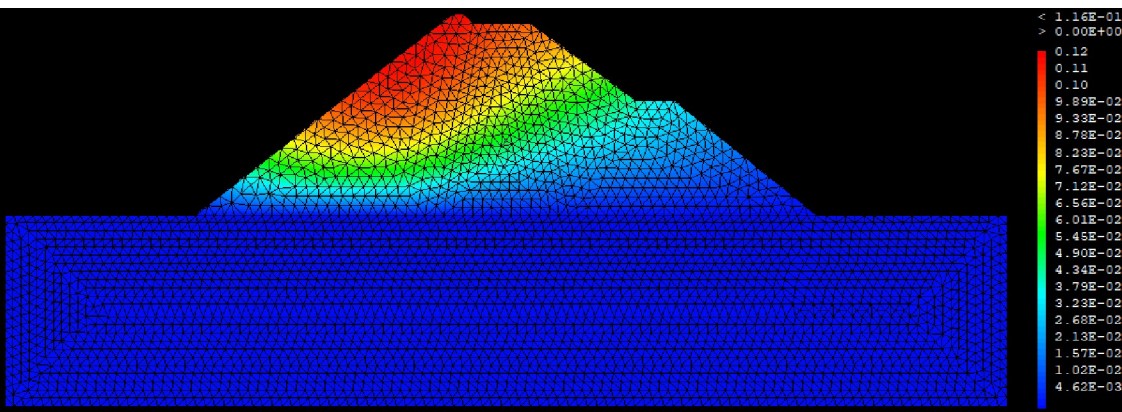

**Fig 29. Horizontal displacements at the water level 528 m obtained by Cast3M software.**

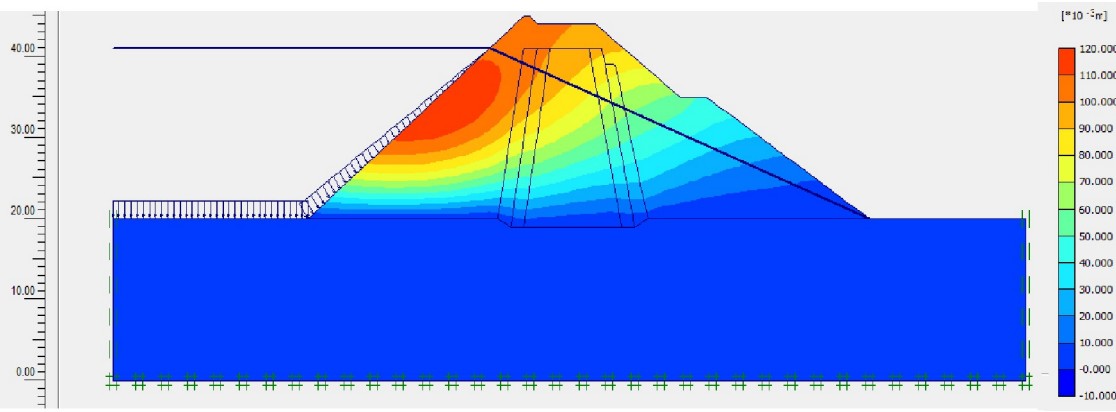

**Fig 30. Horizontal displacements at the water level 528 m obtained by Plaxis software.**

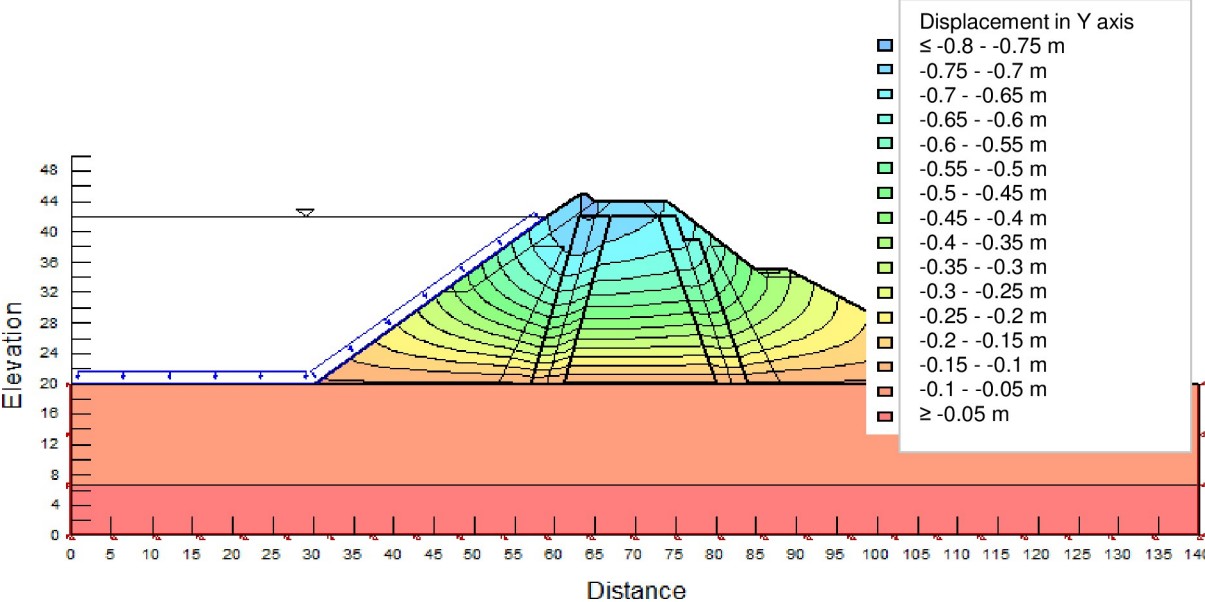

**Fig 31. Vertical displacements at the water level 528 m.**

The maximum total stress at the minimal water level is located at the bottom of the model, and is 912 kPa. At the maximal water level 530 m, the Figs 34 to 39 show the actual behaviour of the earthen dam. The greatest displacements are obtained at the top of the earth dam. The shear failure line of the dam is clearly visible, reflecting the most vulnerable part of the dam subjected to fluctuations in the water level. The maximum horizontal displacement is 20 cm and vertical displacement is 80 cm. The maximum total stress at the minimal water level is located at the bottom of the model, and is 920 kPa. Through these modelling, the results of global displacement according to the level of the reservoir of the dam show that it remains stable as a whole. All these displacements are less than 20 cm. In an advanced monitoring process, we used the auscultation data to have a realistic modelling.

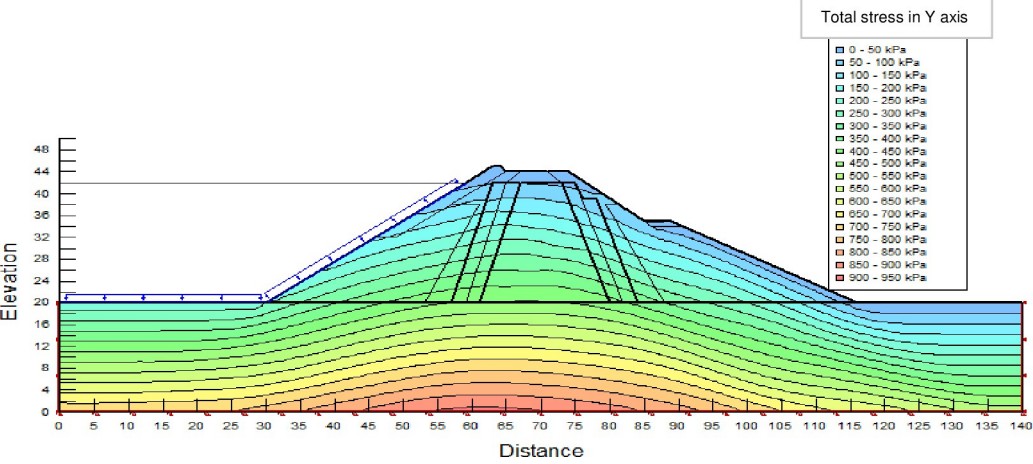

**Fig 32. Total stress (912 kPa at the bottom of the earth dam) at the water level 528 m.**

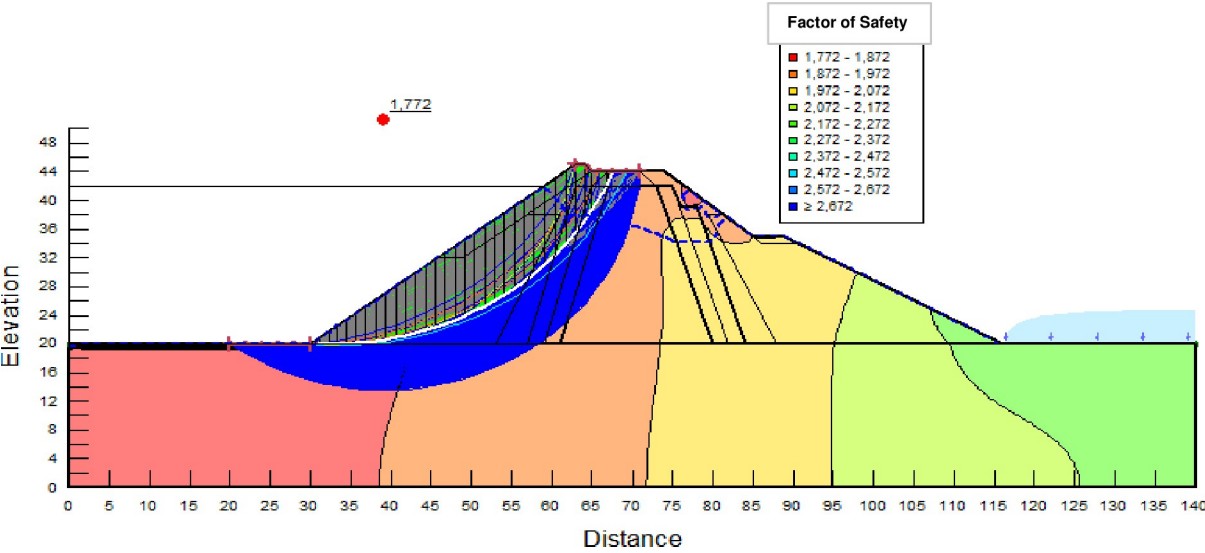

**Fig 33. Critical slope stability of earthen dam of Songloulou at the water level 528 m.**

For the study of cases of earth dams in operation the method exposed in this paper is commonly used [27, 28]. These principles were recently published in American Journal of Construction and Building material [29]. The horizontal displacement results are consistent with those recorded in the direct and indirect pendulums in the middle of the, i.e. 6 to 8 cm. In the growing desire to improve their safety, the integration of numerical investigations in the overall assessment of earthen dam is becoming essential. From the modelling, we have obtained safety factors in slope shear failure greater 1.5 for water levels in the reservoir of 528 and 530 m, reflecting the overall stability of the structure under hydraulic load. Numerical modelling was used to determine the stresses and displacements in the structure. These stresses and displacements make it possible to prevent the failure mechanisms and collapse according to the functional requirements relating to the structure. The iso-values of the vertical stress (Figs 26, 32 and 38) show a concentration of the maximum values at the bottom of the model to the

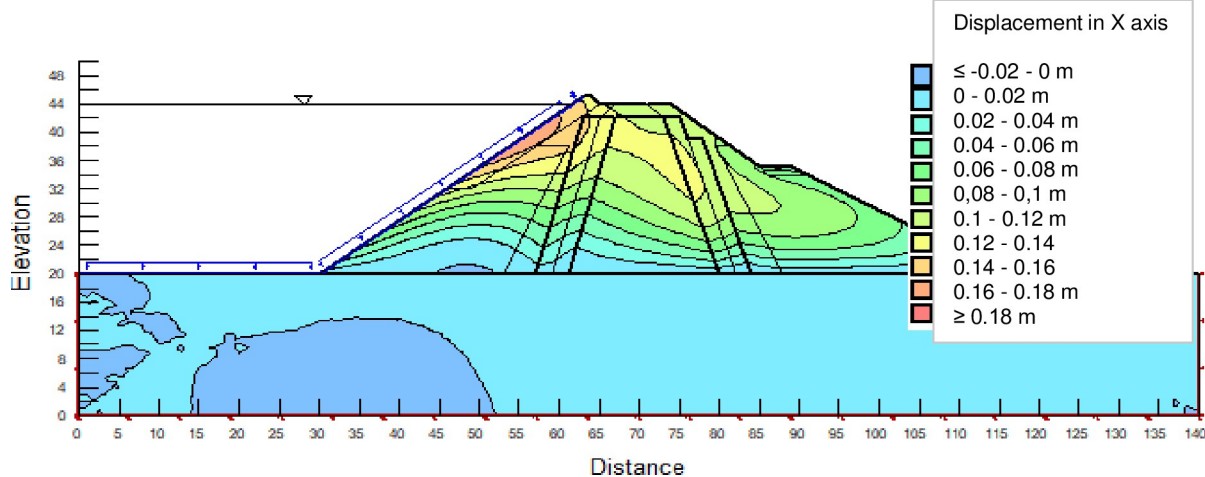

**Fig 34. Horizontal displacements at the water level 530 m obtained by GEOSTUDIO software.**

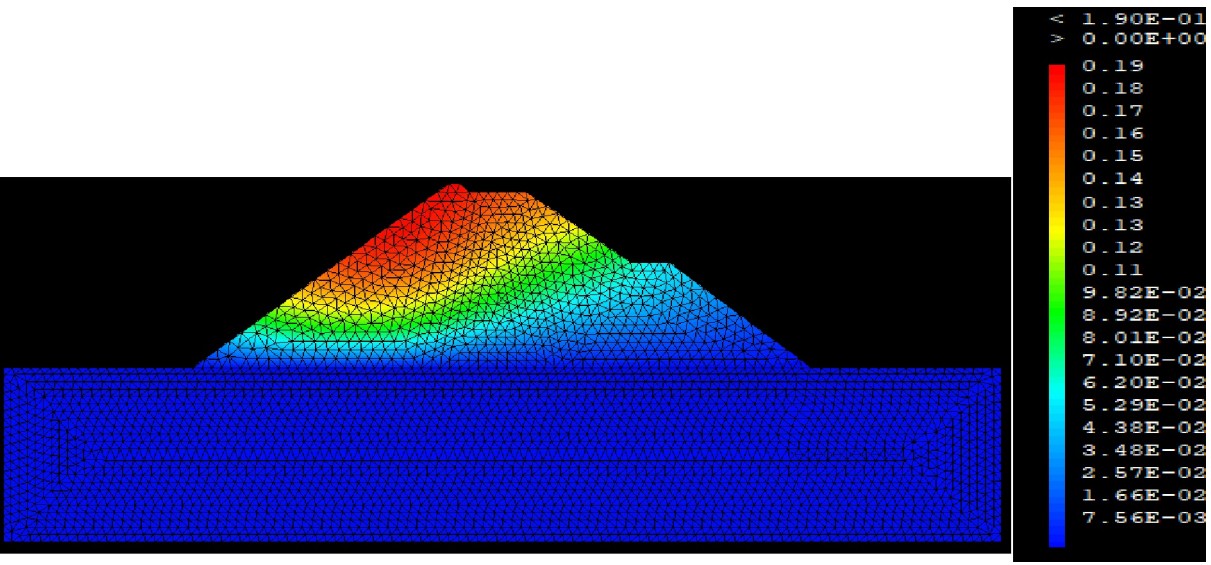

**Fig 35. Horizontal displacements at the water level 530 m obtained by Cast3M software.**

right of the core size. This concentration is consistent with the theoretical distribution mechanism of vertical stresses, $\sigma_{yy} = \Sigma\gamma_i \times h_i$ ($\gamma_i$: unit weight of soil for layer i and $h_i$, thickness of layer i). With parameters from Table 7 and taking into account of the heights of the different layers, the theoretical vertical stress to the right of the core is $\sigma_{yy} = (26 \times 20 + 18 \times 23.5) = 943$ kPa. This value is close to those obtained in Figs 26, 32 and 38.

The iso-values of the vertical displacements along the y-axis are consistent with the vertical stresses: maximum displacement at the top of the model and no displacement at the bottom of the model (Figs 25, 31 and 37) to respect the boundaries conditions of the displacements at the bottom of the model. The iso-values and shape of the horizontal displacements (Figs 24, 28–30, 34–36) are consistent with the sliding failure mechanism of such a structure (circular

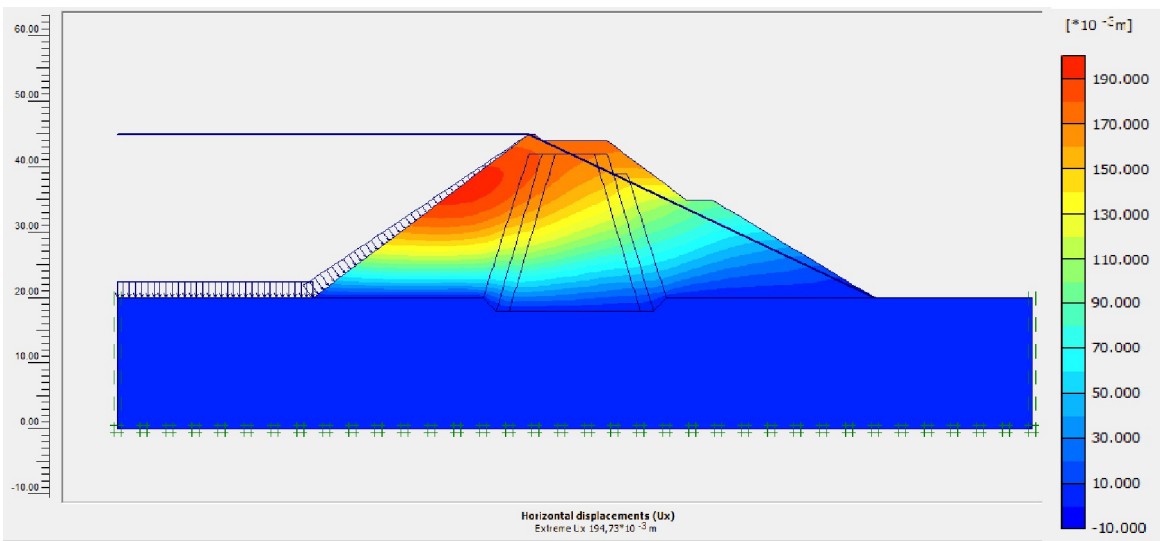

**Fig 36. Horizontal displacements at the water level 530 m obtained by Plaxis software.**

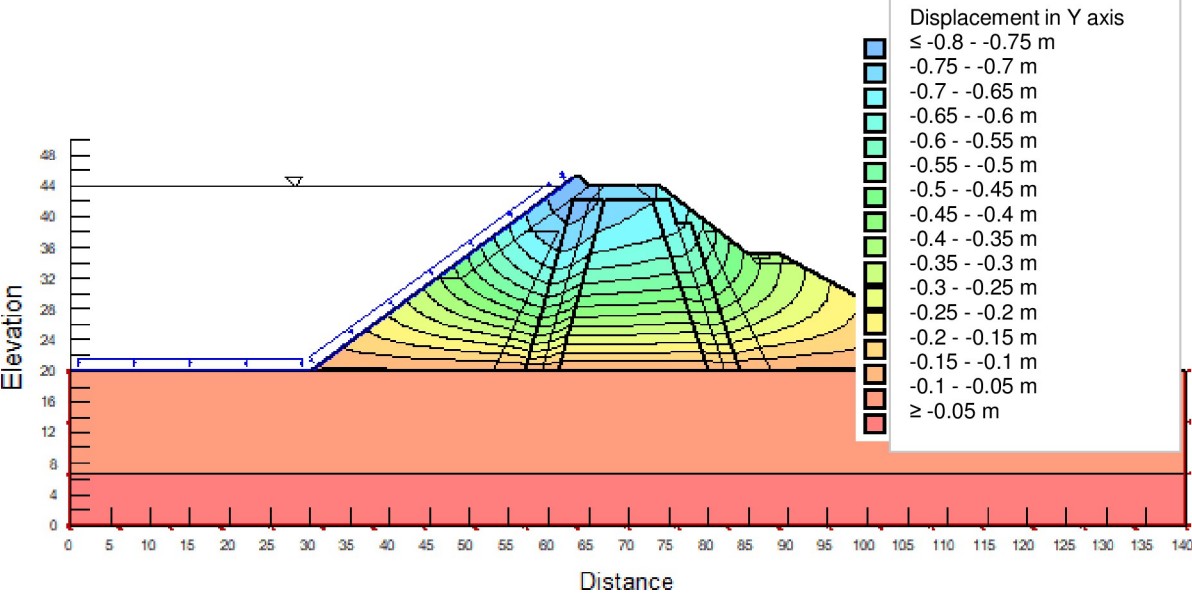

**Fig 37. Vertical displacements at the water level 530 m.**

sliding to the left of the model) (Figs 27, 33 and 39). This sliding mechanism and associated values depend on the level of water retention in the earthen dam. The slip safety factors are greater than 1.5 guarantee the stability of the structure. Nevertheless, precautions should be taken by the managers of this dam when the water level is less than or equal to the 522m rating to ensure its stability. The numerical modelling results found in this paper are consistent with those found by Mouyeaux et al. [15] who did stochastic numerical modelling on an earthen dam in France using the Cast3M calculation code. In his model the iso-values of maximum stresses were at the bottom of the model. The sliding collapse mechanism is circular. In their study, several slip safety factors were evaluated based on the variation in water retention levels in the dam and the mechanical parameters of the materials.

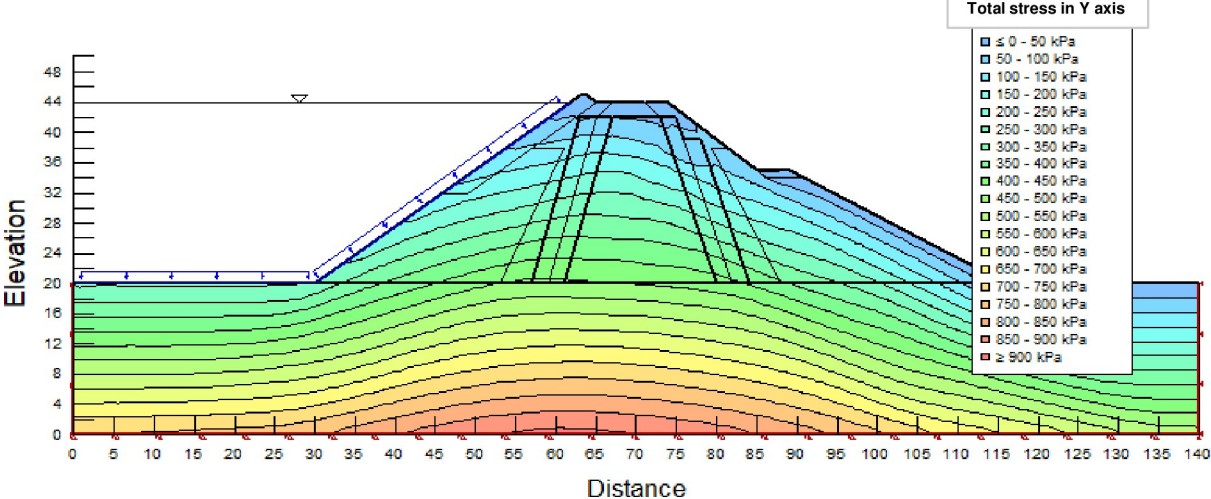

**Fig 38. Total stress at the water level 530 m (920 kPa at the bottom of the earth dam).**

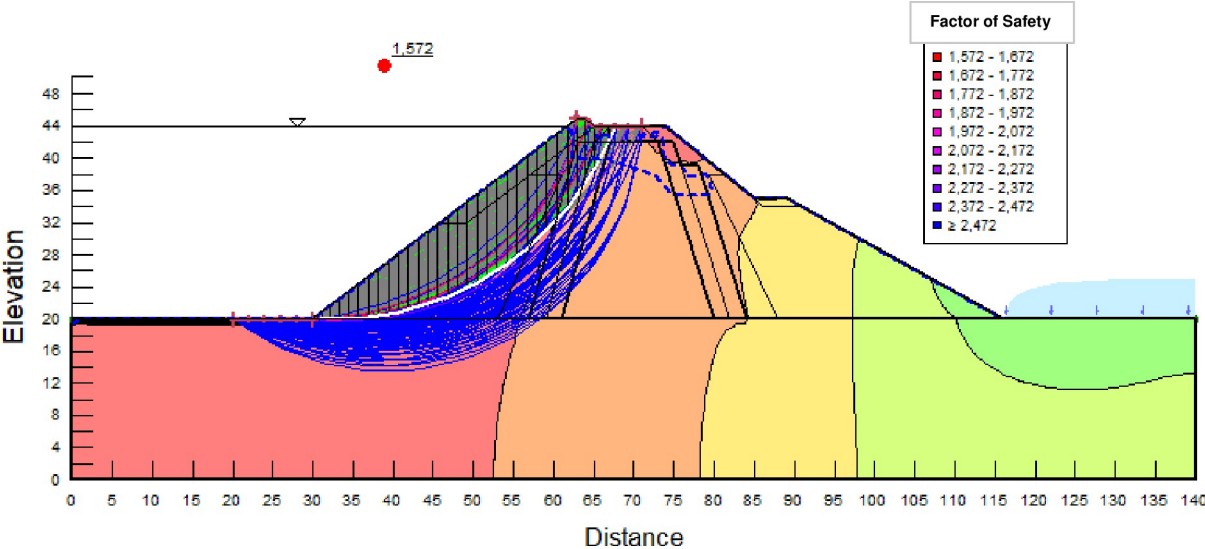

**Fig 39. Critical slope stability of earthen dam of Songloulou at the water level 530 m.**

## 6. Conclusion

In this paper, the prediction qualities of each of the models unequivocally indicate a preference for the ANFIS model. However, when drawing conclusions from such a study, some caution should be exercised. Indeed, all comparisons between the HST and ANFIS model are based on the inseparable data sets from the Songloulou large earthen dam. To validate our model, i.e. to know if it is capable of detecting and anomaly on the dam, it would be necessary to calibrate it using a set of data from a dam shows that the sealing layer separating the upstream part and the downstream part works and protects the downstream from the upstream well, however at the level of the spillway dam there is a significant increase from the piezometric level over time, which shows an increase in the amount of water infiltrated over the years. The safety, security and reliability of dams depend on the lives of thousands of people, so we must be careful not to act hastily in this area. Consequently, it seems to us preferable to model each dam individually and to find in each case an adequate model, for certain structures, the classis model will certainly be more adequate, while for others, the ANFIS model will lead to better results both for the treatment of pressures and displacements. The introduction of the ANFIS model expands the range of models available to perform the necessary control and prediction work on dams. This model also offers an alternative to the classis model when the latter reveals some weaknesses and allows two different models to be used in complementary way. The Anfis method is a learning process that improves over time, which minimizes its error. The HST method is based on an analysis of data in a given time interval; on the other hand, even if

**Table 8. Summary of modelling results and pendulum measurements.**

| Retention level (m) | Ux: Horizontal displacement (cm) | Uy: Vertical displacement (cm) | σyy: Vertical total stress (kPa) | Facteur of safety of slope | Pendulum P1D measurements: Ux (cm) | |
|---|---|---|---|---|---|---|
| | | | | | Upstream -Downstream | left- right |
| 522 | 6 | 74 | 900 | 0.219 | 7.44 | 7.11 |
| 528 | 11.68 | 80 | 912 | 1.772 | | |
| 530 | 20 | 80 | 920 | 1.572 | | |

its estimator is BLUE (Best Linear Unbiased Estimator), the predictions may turn out to be false as soon as the flow conditions change. It is therefore important, before using this method, to carry out a type 1 analysis (Check of data conformity). To predict the overall behaviour of the structure, it necessary to use numerical modelling in order to assess all the inseparable parameters (displacements, pressures, seepages, leak rates). On the GEOSTUDIO modelling, the results of global displacement according to the level of the reservoir of the dam show that it remains stable as a whole. All horizontal displacements are less than 20 cm. In an advanced monitoring process, we used the auscultation data to have a realistic modelling. The horizontal displacement results are consistent with those recorded in the direct and indirect pendulums in the middle of the, i.e. 6 to 8 cm (Table 8). In the growing desire to improve their safety, the integration of numerical investigations in the overall assessment of earthen dam is becoming essential. These investigations make it possible to have the overall behaviour of the structure according various design analysis criteria. From the modelling, we have obtained safety factors in slope shear failure greater 1.5 for water levels in the reservoir of 528 and 530 m, reflecting the overall stability of the structure under hydraulic load. In this paper, an analysis of auscultation devices based on artificial neural networks is implemented. The method adapts well to variations in the parameters influencing the measurement tools and does not require ongoing recalibration during the lifetime of the structure. Unlike the HST method based on data recorded during a period (analysis period), neural networks use the entire data recorded during operation and improve their efficiency over time. The water level of the dam fluctuates between the retention level 522 m and 528 m. The monitoring results and their analyses were backed up by numerical modelling of the earth dam making it possible to have its behaviour according to the mechanical parameters of the materials taken from the geotechnical structure and the level of the hydraulic head.

The horizontal displacements measured on the dam are around 7.4 cm. The average horizontal displacements obtained by numerical modelling at retention level 522 and 528 m is 8.84 cm. These results are consistent with the behaviour of the structure. These results are in agreement with observations on the site. The modelling provides other useful information such as the vertical stresses prevailing at any point in the model. The results of the statistical study and Finite Element Modelling are in agreement with those of recent research throughout the world [10–29, 36–46] for the analysis of the same phenomena. The lack data from altimetric measurements does not make it possible to have settlments or swelling on the top of the dam according to the seasons. This will be the subject of future investigations.

## Acknowledgments

Several people were involved in obtaining the results from this paper; there are very many of them and we cannot name them. Find our warm thanks. The authors acknowledge particularly the ENEO Cameroon, and Geotechnical Laboratories and Engineering's Offices of Cameroon for the facilities and support to perform the experiments.

## Author Contributions

**Conceptualization:** Zoa Ambassa, Jean Chills Amba.

**Data curation:** Zoa Ambassa, Merlin Bodol Momha, Landry Djopkop Kouanang.

**Methodology:** Zoa Ambassa, Jean Chills Amba, Merlin Bodol Momha.

**Project administration:** Jean Chills Amba, Robert Nzengwa.

**Resources:** Pascal Adrien Mbongo.

**Software:** Zoa Ambassa, Merlin Bodol Momha, Landry Djopkop Kouanang.

**Supervision:** Robert Nzengwa.

**Validation:** Zoa Ambassa, Jean Chills Amba, Merlin Bodol Momha.

**Writing – original draft:** Zoa Ambassa, Jean Chills Amba.

**Writing – review & editing:** Zoa Ambassa, Jean Chills Amba.

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
