## [Decision Letter · Decision Letter 0]

10 Apr 2023

PONE-D-23-03305Advanced Monitoring and Numerical modeling of the 

Stability, Safety and Reliability Indicators of the 

Earthen Dam of Songloulou (Cameroon)PLOS ONE

Dear Dr. AMBASSA,

Thank you for submitting your manuscript to PLOS ONE. After careful consideration, we feel that it has merit but does not fully meet PLOS ONE’s publication criteria as it currently stands. Therefore, we invite you to submit a revised version of the manuscript that addresses the points raised during the review process.There are some grammatical mistakes in the manuscript. Please check the English of the manuscript, thoroughly.Figures should come after being cross-referred to in the text. All the figures and tables should be cited in the text.Please provide a main caption for Figure 6.The quality of the figures is poor. The legends are not clear and in some of the figures, the title axis is in French or other languages.In Table 5, please add a title for the D and Da columns that clear the type of these parameters.The detail of the finite element model (section 5) is insufficient. Please provide more details of the FEM.Please provide the definition for the notations used in Table 7.There is no validation or calibration of the FE model. Please elaborate on the accuracy of the model.Please quantify results in the conclusion section and write them point-wise.  Please address the comments specifically and in detail in the response to the reviewer report. Also, do **identify **where in the text you have made the changes by red color, it is particularly helpful to note the page and line numbers from the original manuscript and revision so comparisons can be made.Please submit your revised manuscript by May 25 2023 11:59PM. If you will need more time than this to complete your revisions, please reply to this message or contact the journal office at plosone@plos.org. Please include the following items when submitting your revised manuscript:A rebuttal letter that responds to each point raised by the academic editor and reviewer(s). You should upload this letter as a separate file labeled 'Response to Reviewers'.A marked-up copy of your manuscript that highlights changes made to the original version. You should upload this as a separate file labeled 'Revised Manuscript with Track Changes'.An unmarked version of your revised paper without tracked changes. You should upload this as a separate file labeled 'Manuscript'.

We look forward to receiving your revised manuscript.

Kind regards,

Ahad Javanmardi, Ph.D

Academic Editor

PLOS ONE

“No”

“No”

7. We note that Figure 1 in your submission contain [map/satellite] images which may be copyrighted. All PLOS content is published under the Creative Commons Attribution License (CC BY 4.0), which means that the manuscript, images, and Supporting Information files will be freely available online, and any third party is permitted to access, download, copy, distribute, and use these materials in any way, even commercially, with proper attribution. For these reasons, we cannot publish previously copyrighted maps or satellite images created using proprietary data, such as Google software (Google Maps, Street View, and Earth). For more information, see our copyright guidelines: http://journals.plos.org/plosone/s/licenses-and-copyright.

Reviewers' comments:

Reviewer's Responses to Questions

**Comments to the Author**

1. Is the manuscript technically sound, and do the data support the conclusions?

Reviewer #1: Yes

Reviewer #2: Partly

Reviewer #3: Yes

2. Has the statistical analysis been performed appropriately and rigorously? 

Reviewer #1: Yes

Reviewer #2: Yes

Reviewer #3: Yes

3. Have the authors made all data underlying the findings in their manuscript fully available?

Reviewer #1: Yes

Reviewer #2: No

Reviewer #3: Yes

4. Is the manuscript presented in an intelligible fashion and written in standard English?

Reviewer #1: Yes

Reviewer #2: No

Reviewer #3: Yes

5. Review Comments to the Author

Reviewer #1: For the determination of global stability after advanced long term monitoring, in this paper, we used artificial intelligence for the analysis of water level and displacements data from the Songloulou earth dam of Cameroon. The objective of this paper is to analyze the auscultation data of the Songloulou hydroelectric dam using artificial intelligence and to propose a method of prediction and assistance in the global analysis of said data for the operation of the structure safe. In this paper, the research ideas are clear, the methods are reasonable, and the conclusions are innovative. Therefore, it is recommended to publish it after minor repairs. There are also some aspects that would require further attention. Below the authors can find some remarks to be addressed in the review round:

1，In the introduction, it is suggested to add more references of related research in the recent 3-5 years to highlight the differences and novelty of this study. In addition, the manuscript is pretty much focused on the AI tool while little or marginal context is given regarding the structural application.

2，Specific comparison between ANFIS (Adaptive Neural Fuzzy Inference System) model and other similar models should be given in the introduction. For example, what are the advantages of the ANFIS model in practical engineering applications?

3，The basic model (HST) tends to generalize the system, and approximates the behavior of the piezometer more or less well until 2015 when the peaks become larger and larger, from then on the HST model does not can’t keep up anymore. There are then notable differences between the two models. The reason for this remarkable difference should be explained.

4，In the numerical simulation of the earth dam water level, we should give a comparative analysis with similar research. At the same time, a philosophical explanation of the distribution of stress and displacement should be given.

5，The clarity of the pictures in Figures 25-36 should be improved, which is more convenient for readers to read. For example: increase the font size in all the figures to enhance the readability.

6，What are the lessons learnt from the conclusions other than the observation? Is this finding original to this paper or is it a validation of a previously established notion/fact?

7，The shortcomings and prospects of this study should be given.

Reviewer #2: The current manuscript is not well designed in aspect of English fluency. Due to that, it is very hard to follow the concept of the paper. However, the topic and aim of the paper is still lie in the aim and scope of the journal.

The results are presented in not understanding manner and due to that make the manuscript vague to understand and evaluate.

The results are presented in the form of tables and figures and some doesn't have any justification about the results

Reviewer #3: The authors have made significant contributions to their discipline, and their findings are consistent with the numerical results presented in their paper. Additionally, the authors have made the data underlying the findings in their manuscript available, which makes their work transparent and reproducible.

The manuscript is technically sound, the study was well-designed, and the methods used to collect and analyse the data were appropriate. The authors provided convincing evidence to support their conclusions.

- Please provide the limitations of the work and the future work recommendations in the conclusion section. Highlights areas where further investigation is needed.

- provide an overview for the method section explain the steps of the method before going into the subsections.

- provide a discussion section and only provide the key findings and implications of the study in conclusion section.

6. PLOS authors have the option to publish the peer review history of their article (what does this mean?). If published, this will include your full peer review and any attached files.

Reviewer #1: No

Reviewer #2: **Yes: **Hamed Benisi Ghadim

Reviewer #3: No

---

## [Author Response · Author response to Decision Letter 0]

17 Aug 2023

Dear Referee, 

All modifications have been inserted in the paper in red color

---

## [Decision Letter · Decision Letter 1]

4 Sep 2023

PONE-D-23-03305R1Advanced Monitoring and Numerical modelling of the 

Stability, Safety and Reliability Indicators of the 

Earthen Dam of Songloulou (Cameroon)PLOS ONE

Dear Dr. AMBASSA,

Thank you for submitting your manuscript to PLOS ONE. After careful consideration, we feel that it has merit but does not fully meet PLOS ONE’s publication criteria as it currently stands. Therefore, we invite you to submit a revised version of the manuscript that addresses the points raised during the review process.Please address the comments specifically and in detail in the response to the reviewer report. Also, do **identify** where in the text you have made the changes by red color, it is particularly helpful to note the page and line numbers from the original manuscript and revision so comparisons can be made. Please submit your revised manuscript by Oct 19 2023 11:59PM. If you will need more time than this to complete your revisions, please reply to this message or contact the journal office at plosone@plos.org. Please include the following items when submitting your revised manuscript:A rebuttal letter that responds to each point raised by the academic editor and reviewer(s). You should upload this letter as a separate file labeled 'Response to Reviewers'.A marked-up copy of your manuscript that highlights changes made to the original version. You should upload this as a separate file labeled 'Revised Manuscript with Track Changes'.An unmarked version of your revised paper without tracked changes. You should upload this as a separate file labeled 'Manuscript'.If applicable, we recommend that you deposit your laboratory protocols in protocols.io to enhance the reproducibility of your results. Protocols.io assigns your protocol its own identifier (DOI) so that it can be cited independently in the future. For instructions see: https://journals.plos.org/plosone/s/submission-guidelines#loc-laboratory-protocols. Additionally, PLOS ONE offers an option for publishing peer-reviewed Lab Protocol articles, which describe protocols hosted on protocols.io. Read more information on sharing protocols at https://plos.org/protocols?utm_medium=editorial-email&utm_source=authorletters&utm_campaign=protocols.

We look forward to receiving your revised manuscript.

Kind regards,

Ahad Javanmardi, Ph.D

Academic Editor

PLOS ONE

Journal Requirements:

Reviewers' comments:

Reviewer's Responses to Questions

**Comments to the Author**

1. If the authors have adequately addressed your comments raised in a previous round of review and you feel that this manuscript is now acceptable for publication, you may indicate that here to bypass the “Comments to the Author” section, enter your conflict of interest statement in the “Confidential to Editor” section, and submit your "Accept" recommendation.

Reviewer #1: (No Response)

Reviewer #2: All comments have been addressed

2. Is the manuscript technically sound, and do the data support the conclusions?

Reviewer #1: Yes

Reviewer #2: Yes

3. Has the statistical analysis been performed appropriately and rigorously? 

Reviewer #1: Yes

Reviewer #2: Yes

4. Have the authors made all data underlying the findings in their manuscript fully available?

Reviewer #1: Yes

Reviewer #2: Yes

5. Is the manuscript presented in an intelligible fashion and written in standard English?

Reviewer #1: Yes

Reviewer #2: Yes

6. Review Comments to the Author

Reviewer #1: The author partially solved the problems raised by the reviewers, which is worthy of recognition. However, there are still some important issues in the manuscript that have not been fully explained. The current manuscript form is suggested to be slightly revised before it can be published by PLOS ONE. The following important problems should be fully solved. 1，The basic model (HST) tends to generalize the system, and approximates the behavior of the piezometer more or less well until 2015 when the peaks become larger and larger, from then on the HST model does not can’t keep up anymore. There are then notable differences between the two models. The reason for this remarkable difference should be explained.

2，In the numerical simulation of the earth dam water level, we should give a comparative analysis with similar research. At the same time, a philosophical explanation of the distribution of stress and displacement should be given.

3，The clarity of the pictures in Figures 25-36 should be improved, which is more convenient for readers to read. For example: increase the font size in all the figures to enhance the readability.

4，What are the lessons learnt from the conclusions other than the observation? Is this finding original to this paper or is it a validation of a previously established notion/fact?

Reviewer #2: Thank you for all your revisions and I believe this manuscript at this format has potential to be published in the PLOS ONE journal.

7. PLOS authors have the option to publish the peer review history of their article (what does this mean?). If published, this will include your full peer review and any attached files.

Reviewer #1: No

Reviewer #2: **Yes: **Hamed Benisi Ghadim

---

## [Author Response · Author response to Decision Letter 1]

12 Sep 2023

All modifications have been inserted in the red color in the file Revised Manuscript withTrack Changes

---

## [Editor Report · Decision Letter 2]

14 Sep 2023

PONE-D-23-03305R2Advanced Monitoring and Numerical modelling of the 

Stability, Safety and Reliability Indicators of the 

Earthen Dam of Songloulou (Cameroon)PLOS ONE

Dear Dr. AMBASSA,

Thank you for submitting your manuscript to PLOS ONE. After careful consideration, we feel that it has merit but does not fully meet PLOS ONE’s publication criteria as it currently stands. Therefore, we invite you to submit a revised version of the manuscript that addresses the points raised during the review process.

 **Authors failed to answer Reviwer 1. Please answer Reviewer 1 comments for the first revision. **Please address the comments specifically and in detail in the response to the reviewer report. Also, do **identify** where in the text you have made the changes by red color, it is particularly helpful to note the page and line numbers from the original manuscript and revision so comparisons can be made.Please submit your revised manuscript by Oct 29 2023 11:59PM. If you will need more time than this to complete your revisions, please reply to this message or contact the journal office at plosone@plos.org. Please include the following items when submitting your revised manuscript:A rebuttal letter that responds to each point raised by the academic editor and reviewer(s). You should upload this letter as a separate file labeled 'Response to Reviewers'.A marked-up copy of your manuscript that highlights changes made to the original version. You should upload this as a separate file labeled 'Revised Manuscript with Track Changes'.An unmarked version of your revised paper without tracked changes. You should upload this as a separate file labeled 'Manuscript'.If applicable, we recommend that you deposit your laboratory protocols in protocols.io to enhance the reproducibility of your results. Protocols.io assigns your protocol its own identifier (DOI) so that it can be cited independently in the future. For instructions see: https://journals.plos.org/plosone/s/submission-guidelines#loc-laboratory-protocols. Additionally, PLOS ONE offers an option for publishing peer-reviewed Lab Protocol articles, which describe protocols hosted on protocols.io. Read more information on sharing protocols at https://plos.org/protocols?utm_medium=editorial-email&utm_source=authorletters&utm_campaign=protocols.

We look forward to receiving your revised manuscript.

Kind regards,

Ahad Javanmardi, Ph.D

Academic Editor

PLOS ONE
---

## [Author Response · Author response to Decision Letter 2]

19 Sep 2023

All modification has been inserted in the paper in red color in the file labeled ‘Revised Manuscript with Track Changes’.

---

## [Decision Letter · Decision Letter 3]

29 Sep 2023

Advanced Monitoring and Numerical modelling of the 

Stability, Safety and Reliability Indicators of the 

Earthen Dam of Songloulou (Cameroon)

PONE-D-23-03305R3

Dear Dr. AMBASSA,

We’re pleased to inform you that your manuscript has been judged scientifically suitable for publication and will be formally accepted for publication once it meets all outstanding technical requirements.

Kind regards,

Ahad Javanmardi, Ph.D

Academic Editor

PLOS ONE

Reviewers' comments:

Reviewer's Responses to Questions

**Comments to the Author**

1. If the authors have adequately addressed your comments raised in a previous round of review and you feel that this manuscript is now acceptable for publication, you may indicate that here to bypass the “Comments to the Author” section, enter your conflict of interest statement in the “Confidential to Editor” section, and submit your "Accept" recommendation.

Reviewer #1: All comments have been addressed

2. Is the manuscript technically sound, and do the data support the conclusions?

Reviewer #1: Yes

3. Has the statistical analysis been performed appropriately and rigorously? 

Reviewer #1: Yes

4. Have the authors made all data underlying the findings in their manuscript fully available?

Reviewer #1: Yes

5. Is the manuscript presented in an intelligible fashion and written in standard English?

Reviewer #1: Yes

6. Review Comments to the Author

Reviewer #1: The authors have addressed my comments in a satisfactory way. It now can be considered for publication in PLOS ONE.

7. PLOS authors have the option to publish the peer review history of their article (what does this mean?). If published, this will include your full peer review and any attached files.

Reviewer #1: No

---

## [Editor Report · Acceptance letter]

3 Oct 2023

PONE-D-23-03305R3 

Advanced Monitoring and Numerical modelling of the Stability, Safety and Reliability Indicators of the Earthen Dam of Songloulou (Cameroon) 

Dear Dr. Ambassa:

I'm pleased to inform you that your manuscript has been deemed suitable for publication in PLOS ONE. Congratulations! Your manuscript is now with our production department. 

Kind regards, 

on behalf of

Associate Professor Ahad Javanmardi 

Academic Editor

PLOS ONE